# The transcriptional landscape of Venezuelan equine encephalitis virus (TC-83) infection

Zhiyuan Yao[1☯], Fabio Zanini[2,3☯]*, Sathish Kumar[1], Marwah Karim[1], Sirle Saul[1], Nishank Bhalla[4], Nuttada Panpradist[1,5], Avery Muniz[1], Aarthi Narayanan[4], Stephen R. Quake[2,6,7], Shirit Einav[1,8]*

**1** Division of Infectious Diseases and Geographic Medicine, Department of Medicine, Stanford University School of Medicine, Stanford, California, United States of America, **2** Department of Bioengineering, Stanford University, Stanford, California, United States of America, **3** Lowy Cancer Research Centre, University of New South Wales, Sydney, Australia, **4** National Center for Biodefence and Infectious Disease, Biomedical Research Laboratory, School of Systems Biology, George Mason University, Manassas, Virginia, United States of America, **5** Department of Bioengineering, University of Washington, Seattle, Washington, United States of America, **6** Chan Zuckerberg Biohub, San Francisco, California, United States of America, **7** Department of Applied Physics, Stanford University, Stanford, California, United States of America, **8** Department of Microbiology and Immunology, Stanford University School of Medicine, Stanford, California, United States of America

☯ These authors contributed equally to this work.
* fabio.zanini@unsw.edu.au (FZ); seinav@stanford.edu (SE)

**Data Availability Statement:** In line with the spirit of open science at PNTD, we have posted this manuscript in BioRxiv. We have also provided all data, code, and analyses for public scrutiny on

## Abstract

Venezuelan Equine Encephalitis Virus (VEEV) is a major biothreat agent that naturally causes outbreaks in humans and horses particularly in tropical areas of the western hemisphere, for which no antiviral therapy is currently available. The host response to VEEV and the cellular factors this alphavirus hijacks to support its effective replication or evade cellular immune responses are largely uncharacterized. We have previously demonstrated tremendous cell-to-cell heterogeneity in viral RNA (vRNA) and cellular transcript levels during flaviviral infection using a novel virus-inclusive single-cell RNA-Seq approach. Here, we used this unbiased, genome-wide approach to simultaneously profile the host transcriptome and vRNA in thousands of single cells during infection of human astrocytes with the live-attenuated vaccine strain of VEEV (TC-83). Host transcription was profoundly suppressed, yet "superproducer cells" with extremely high vRNA abundance emerged during the first viral life cycle and demonstrated an altered transcriptome relative to both uninfected cells and cells with high vRNA abundance harvested at later time points. Additionally, cells with increased structural-to-nonstructural transcript ratio exhibited upregulation of intracellular membrane trafficking genes at later time points. Loss- and gain-of-function experiments confirmed pro- and antiviral activities in both vaccine and virulent VEEV infections among the products of transcripts that positively or negatively correlated with vRNA abundance, respectively. Lastly, comparison with single cell transcriptomic data from other viruses highlighted common and unique pathways perturbed by infection across evolutionary scales. This study provides a high-resolution characterization of the VEEV (TC-83)-host interplay, identifies candidate targets for antivirals, and establishes a comparative single-cell approach to study the evolution of virus-host interactions.

GitHub at https://github.com/saberyzy/VEEV-single_cell. The single cell RNA-Seq data for this study is available on GEO at submission number: GSE145815 (https://www.ncbi.nlm.nih.gov/geo/query/acc.cgi?acc=GSE145815). Processed count and metadata tables are also available on FigShare at https://figshare.com/articles/Untitled_Item/11874198.

**Funding:** This work was supported by HDTRA11810039 from the Defense Threat Reduction Agency (DTRA)/Fundamental Research to Counter Weapons of Mass Destruction to SE and AN, by the Chan Zuckerberg Biohub to SQ, and by a Stanford Bio-X Interdisciplinary Initiative Program Award to SE. ZY was supported by the Maternal and Child Health Research Institute, Lucile Packard Foundation for Children's Health). NP was supported by the University of Washington School of Medicine Guy Tribble and Susan Barnes Graduate Discovery Fellowship. We thank investigators who have provided plasmids (see Methods). The funders had no role in study design, data collection and analysis, decision to publish, or preparation of the manuscript.

**Competing interests:** The authors have declared that no competing interests exist.

## Author summary

Little is known about the host response to Venezuelan Equine Encephalitis Virus (VEEV) and the cellular factors this alphavirus hijacks to support effective replication or evade cellular immune responses. Monitoring dynamics of host and viral RNA (vRNA) during viral infection at a single-cell level can provide insight into the virus-host interplay at a high resolution. Here, a single-cell RNA sequencing technology that detects host and viral RNA was used to investigate the interactions between TC-83, the vaccine strain of VEEV, and the human host during the course of infection of U-87 MG cells (human astrocytoma). Virus abundance and host transcriptome were heterogeneous across cells from the same culture. Subsets of differentially expressed genes, positively or negatively correlating with vRNA abundance, were identified and subsequently *in vitro* validated as candidate proviral and antiviral factors, respectively, in TC-83 and/or virulent VEEV infections. In the first replication cycle, "superproducer" cells exhibited rapid increase in vRNA abundance and unique gene expression patterns. At later time points, cells with increased structural-to-nonstructural transcript ratio demonstrated upregulation of intracellular membrane trafficking genes. Lastly, comparing the VEEV dataset with published datasets on other RNA viruses revealed unique and overlapping responses across viral clades. Overall, this study improves the understanding of VEEV-host interactions, reveals candidate targets for antiviral approaches, and establishes a comparative single-cell approach to study the evolution of virus-host interactions.

## Introduction

For more than a century, Venezuelan Equine Encephalitis Virus (VEEV), a member of the *Alphavirus* genus, has been the causative agent of outbreaks of febrile neurological disease in both animals and humans in Central and South America [1,2]. The incidence of VEEV infection is underestimated since early symptoms are non-specific [2]. While typically transmitted via a mosquito bite, VEEV is also infectious as an aerosol, hence it is considered a major bioterrorism threat [3]. To date, no US FDA approved drugs or vaccines against VEEV are available. A deeper understanding of VEEV biology in human cells is required to advance the development of effective countermeasures against VEEV.

Because VEEV is a biosafety level 3 pathogen, TC-83, a live-attenuated vaccine strain, is commonly used for research purposes [4]. Although attenuated, VEEV TC-83 replicates rapidly: viral protein production is observed as early as 6 hours postinfection (hpi) in human astrocytoma cells (U-87 MG) at multiplicity of infection (MOI) of 2, and over $10^{10}$ copies of intracellular viral RNA (vRNA) can be detected by 24 hpi [5]. It remains unknown, however, whether a large number of cells, each producing a small number of virions, or a few "superproducer" cells drive this effective virus production. Productive replication is associated with profound shutdown of host gene transcription [6]. Nevertheless, since the virus relies on cellular machineries, it is important to identify which host factors are "spared" from this shutdown, as they may represent essential factors for effective viral replication.

The genome of VEEV is an ~11.5 kb single-stranded positive-sense RNA. The genomic RNA contains two regions. The 5' two-thirds of the genome constitutes the first open reading frame (ORF), which encodes the nonstructural (ns) proteins required for viral RNA synthesis (nsP1-4). The 3' one-third of the genome encodes the structural proteins. The structural proteins (capsid, envelope glycoproteins E1-3, 6k, and transframe (TF) protein) are translated

from a second ORF that is expressed through the production of a subgenomic mRNA from an internal promoter in the negative-strand RNA replication intermediate and function in the assembly of new virions and their attachment and entry into cells [7]. While the stoichiometry of the genomic and subgenomic transcripts in the setting of VEEV infection has not been characterized, the transcription of the subgenomic RNA of a related alphavirus, Sindbis virus (SINV), was shown to be ~3-fold higher than the genomic RNA during late stages of the viral lifecycle [8,9], supporting a switch towards increased synthesis of structural proteins required for virion formation over nonstructural proteins required primarily for viral RNA replication [10,11].

The understanding of the alphavirus life cycle is largely based on studies conducted with the non-pathogenic SINV and Semliki forest virus (SFV). Alphaviruses enter their target cells via clathrin-mediated endocytosis and release their nucleocapsid into the cytoplasm via fusion with endosomal membranes, followed by translation and processing of the nonstructural polyprotein [12]. Viral RNA replication occurs within membrane invaginations called spherules that are thought to be derived from the plasma membrane, endoplasmic reticulum and late endosomes and are subsequently incorporated into type 1 cytopathic vacuoles (CPV)-I composed of modified endosomes and lysosomes [13–16]. Production of genomic RNA and subsequently subgenomic RNA are followed by polyprotein translation and processing. The current model of infectious alphavirus production suggests that the genomic RNA is packaged by the capsid in the cytoplasm, and that the viral glycoproteins traffic via membrane structures, presumed to be *trans*Golgi-derived (CPV-II), to budding sites on the plasma membrane, followed by membrane curving and scission, facilitating envelopment of the nucleocapsid [16–18].

Although VEEV is predicted to extensively interact with cellular factors to effectively replicate and evade cellular immune responses, like other RNA viruses, little is known about these interactions. A recent small interfering RNA (siRNA) screen revealed a requirement for actin-remodeling pathway proteins including ARF1, RAC1, PIP5K1-α, and ARP3 in VEEV infection and specifically in promoting viral glycoprotein transport to the plasma membrane [19]. Various other cellular proteins, such as DDX-1 and -3 [20], have been reported to interact with viral proteins and have proviral functions. The transcript levels of antiviral factors including IFITM3 [21] and members of the PARP protein family [22] were shown to be upregulated in VEEV infection via genome-wide microarray screenings. Nevertheless, to the best of our knowledge, the interplay between VEEV and the human host has not been studied to date via an unbiased, single cell genome-wide approach.

Single cell RNA sequencing (scRNA-Seq) has demonstrated utility for understanding the heterogeneity of both viral and cellular transcriptome dynamics at a high resolution. We have recently developed virus-inclusive single-cell RNA-Seq (viscRNA-Seq), an approach to simultaneously profile host and viral gene expression in thousands of single cells [23]. The studies we and others have conducted in cell lines infected with dengue (DENV), Zika (ZIKV), influenza A (IAV) [24,25] and West Nile (WNV) viruses [26] and our results in samples from DENV-infected patients [27] revealed a tremendous cell-to-cell heterogeneity in both vRNA abundance and levels of host factors that support or restrict infection. Moreover, we have demonstrated the utility of this approach in identifying novel cellular factors that support or restrict viral infection [23]. We have therefore hypothesized that studying VEEV-TC-83 transcriptome dynamics at a single cell resolution may overcome challenges related to the high viral replication rate, thereby highlighting specific transcriptomic signatures above the suppressed transcriptional landscape and identifying candidate cellular factors that may support or restrict VEEV replication.

We conducted a longitudinal study of virus-host cell interactions across 24 hours of VEEV-TC-83 infection in U-87 MG cells via viscRNA-Seq. We detected extreme heterogeneity

in vRNA abundance and host transcriptome across cells from the same culture. To overcome the challenge presented by this uneven and rapid viral replication, we stratified cell populations based on vRNA abundance rather than time postinfection and correlated cellular gene expression with both (i) total vRNA and (ii) the ratio of total (genomic + subgenomic) to genomic vRNA. These approaches enabled identification of genes whose expression is altered during VEEV-TC-83 infection, several of which were then confirmed via loss-of-function and gain-of-function experiments in cells infected with the vaccine or virulent VEEV strains as candidate pro- and antiviral factors, respectively. Moreover, we revealed a small population of "superproducer cells" that drives the rapid increase in vRNA in the first replication cycle and a cell population that harbors excess of the structural over nonstructural viral ORFs at late stages of viral infection, both associated with distinct host gene expression patterns. Lastly, comparison of the VEEV dataset with published data on other RNA viruses revealed unique and overlapping host gene responses across viral clades, highlighting the utility of comparative single-cell transcriptomics.

## Materials and methods

### Cells

U-87 MG, BHK-21 (baby hamster kidney) and Vero (African green monkey kidney epithelial) cell lines were obtained from ATCC (Manassas, VA). Cells were grown in Dulbecco's Modified Eagle's medium (DMEM, Mediatech, Manassas, VA), supplemented with 1% Penicillin-Streptomycin solution, 1% L-glutamine 200 mM (Thermo Fisher Scientific, Waltham, MA) and 10% Fetal Bovine Serum (FBS, Omega Scientific, INC, Tarzana, CA). Cells were maintained in a humidified incubator with 5% CO2 at 37˚C. Cells were tested negative for mycoplasma by the MycoAlert mycoplasma detection kit (Lonza, Morristown, NJ).

### Plasmids and virus constructs

The plasmids encoding infectious VEEV-TC-83 with a GFP reporter (VEEV-TC-83-Cap-eGFP-Tav, hereafter VEEV-TC-83-GFP) or a nanoluciferase reporter (VEEV TC-83-Cap-nLuc-Tav, hereafter VEEV-TC-83-nLuc) were a gift from Dr. William B. Klimstra (Department of Immunology, University of Pittsburgh) [28]. Open reading frames (ORFs) encoding 11 hits were selected from the Human ORFeome library of cDNA clones (Open Biosystems) [29] and recombined into a FLAG (for FLAG tagging) vector using Gateway technology (Invitrogen).

### Virus production

Viral RNA (vRNA) (VEEV-TC-83-GFP or nLuc) was transcribed *in vitro* from cDNA plasmid templates linearized with MluI via MegaScript Sp6 kit (Invitrogen #AM1330) and electroporated into BHK-21 cells. VEEV was harvested from the supernatant 24 hours postelectroporation, clarified from cell debris by centrifugation, and stored at -80˚C. The non-reporter, wild type VEEV-TC-83, live attenuated strain, and the wild type Trinidad Donkey (TrD) strain were obtained from BEI Resources. All VEEV-TrD experiments were performed under BSL3 conditions. Virus stock titers were determined by standard plaque assay on Vero cells, and titers were expressed as plaque forming units/ml (PFU/ml).

### Infection assays

U-87 MG cells were infected with VEEV-TC-83-GFP at various MOIs (0, 0.1, and 1) and harvested at distinct time points postinfection. For the functional screens, U-87 MG cells were

infected with either VEEV-TC-83-nLuc in 8 replicates (MOI = 0.01), non-reporter VEEV-TC-83, or wild type VEEV TrD in triplicates (MOI = 0.001). Overall infection was measured at 18 hpi via a nanoluciferase assay using a luciferin solution obtained from the hydrolysis of its O-acetylated precursor, hikarazine-103 (prepared by Dr. Yves Janin, Pasteur Institute, France) as a substrate [30,31] or at 24 hpi via standard plaque assays (viral titers in the control samples in these experiments were $> 10^8$ PFU/ml).

## Detection of infected cells using VEEV-specific capture oligo

To optimize the viscRNA-Seq protocol for a wide dynamic range of vRNA amount per VEEV-infected cells, we designed and screened eight capture oligonucleotides (**S1 Table**).

To screen these capture oligos, we first generated cDNA from VEEV-infected cells in the presence of each or combinations of VEEV-specific capture oligo. Specifically, 30 pg of both vRNA and cellular RNA purified from VEEV-infected cells was reverse-transcribed to cDNA in a reaction containing SuperScript IV reverse transcriptase, 1X First Strand buffer (Invitrogen), 5 mM DTT, 1 M betaine, 6 mM $MgCl_2$, 1 μM oligo dT and each or combinations of 100 nM reverse VEEV capture oligo. Subsequently, cDNA underwent 21-cycle PCR amplification using ISPCR primers. cDNA was then purified using Ampure XP beads (Beckman Coulter) at the ratio of 0.8 and eluted in 15 μL EB buffer. Fragments of purified, concentrated cDNA were visualized and quantified using bioanalyzer (DNA High Sensitivity kit, Agilent Technologies). To quantify the amount of vRNA captured by each or combinations of capture oligos, these purified cDNA were also subjected to qPCR (Hot-start OneTaq (New England Biolabs), 1x Standard Taq buffer, 1x Evagreen (Biotium), forward primer: ATTCTAAGCACAAGTAT-CATTGTAT and reverse primer: TTAGTTGCATACTTATACAATCTGT located upstream of all the capture oligos. VEEV_1 and VEEV_2 yielded the highest copies of viral cDNA and did not generate significant primer dimers. Therefore, this combination of the capture oligo was selected for downstream experiments.

## Single cell sorting

At each time point, cells were trypsinized for 10 min, spun and resuspended in 1 mL fresh media. Within 15 min, cells were pelleted again and resuspended in 2 ml 1X phosphate-buffered saline (PBS) buffer at a concentration of $10^6$ cells/ml. Cells were filtered through a 40 μm filter into a 5 ml FACS tube and sorted on a Sony SH800 sorter using SYTOX Blue dead cell stain (ThermoFisher) to distinguish living cells from dead cells and debris. VEEV harboring cells were sorted based on GFP signal. Cells were sorted into 384-well PCR plates containing 0.5 μl of lysis buffer using 'Single cell' purity mode. A total of 12 384-well plates of single cells were sorted for the VEEV time course.

## Lysis buffer, reverse transcription, and PCR

To capture and amplify both mRNA and vRNA from the same cell, the Smart-seq2 protocol was adapted (Picelli et al., 2014). All volumes were reduced by a factor of 12 compared to the original protocol to enable high-throughput processing of 384-well plates. External RNA Controls Consortium (ERCC) spike-in RNA was added at a concentration of 1:10 of the normal amount. The lysis buffer contained 100 nM of oligo-dT primer, 100 nM of virus specific capture oligo mix (i.e. VEEV_1 and VEEV_2) to capture the positive-stranded virus RNA.

Other virus-specific primers and higher primer concentrations were tested but resulted in a large fraction of primer dimers. In order to reduce interference between the virus-specific primer and the Template Switching Oligo (TSO) used to extend the reverse transcription (RT) products, a 5'-blocked biotinylated TSO was used at the standard concentration. RT and PCR

of the cDNA were performed in a total volume of 1 μl and 2.5 μl for each well respectively. The resulting cDNAs were amplified for 21 cycles. Lambda exonuclease was added to the PCR buffer at a final concentration of 0.0225 U/μl and the RT products were incubated at 37°C for 30 min before melting the RNA-DNA hybrid (as it was observed that this reduced the amount of low-molecular weight bands from the PCR products). The cDNA was then diluted 1 to 7 in EB buffer for a final volume of 17.5 μl. All pipetting steps were performed using a Mosquito HTS robotic platform (TTP Labtech).

## cDNA quantification

To quantify the amount of cDNA in each well after PCR, a commercial fluorometric assay was used (ThermoFisher Quant-It Picogreen). Briefly, 1 μl of cDNA and 50 μl of 1:200 dye-buffer mix were pipetted together into a flat-bottom 384-well plate (Corning 3711). For each plate, six wells were used as standard wells. 1 μl dd $H_2O$ was added into one standard well as blank. The standard solutions were diluted into 5 concentrations (0.1, 0.2, 0.4, 0.8, 1.6 ng/μl) and added 1μl into the remaining 5 standard wells. The plate was vortexed for 2 min, centrifuged, incubated in the dark for 5 min, and measured on a plate reader at wavelength 550 nm. cDNA concentrations were calculated via an affine fit to the standard wells.

## Library preparation and sequencing

For each time point, one plate was sent for library preparation and sequencing. In total, 6 plates (2304 cells) were prepared. Sequencing libraries were prepared using the illumina Nextera XT kit according to the manufacturer's instructions, with the following exceptions: (1) we used a smaller reaction volume (around 1 μl per cell); (2) we chose a slightly higher cDNA concentration (0.4 ng/μl) as input, to compensate for the lack of bead purification upstream; (3) we used the commercial 24 i7 barcodes and the 64 new i5 barcode sequences. We noticed a low level of cross-talk between these barcodes, indicated by up to five virus reads found in a few uninfected cells. However, considering that a sizeable fraction of cells in the same sequencing run (late infected and high MOI) had thousands of virus reads, the amount of cross-talk between barcodes appears to be of the order of 1 in 10,000 or less. We used Illumina Novaseq sequencer for sequencing.

## Bioinformatics pipeline

Sequencing reads were mapped against the human GRCh38 genome with supplementary ERCC sequences and TC-83-VEEV-GFP genome using STAR Aligner [32]. Genes were counted using htseq-count [33]. The Stanford high performance computing cluster Sherlock 2.0 was used for the computations. Once the gene/virus counts were available, the downstream analysis was performed on laptops using the packages Seurat [34] and singlet (https://github.com/iosonofabio/singlet), as well as custom R and Python scripts. Ggplot2 [35], matplotlib [36] and seaborn [37] were used for plotting.

For the mutational analysis, all reads mapping to VEEV were extracted from all cells with a unique identifier of the cell of origin, and all four possible alleles at each nucleotide were counted by custom scripts based on pysam (https://github.com/pysam-developers/pysam) and wrapped in an xarray Dataset [38]. The analysis was restricted to infected cells with an average of 100 or more reads per viral genomic site to reduce shot noise.

Comparison with flaviviruses was performed as follows. First, host genes with similar expression (within a factor of 10) in counts per millions (cpm) were identified. Within that class, correlations with vRNA for VEEV, DENV, ZIKV were computed separately. Host factors with the highest discrepancies between pairs of viruses were identified. For Fig 5A–5C, a gene

was chosen from the most discrepant genes exemplifying the different behaviors observed and the cells were scattered using vRNA abundance and gene expression axes, and colored by virus. For Fig 5D, the host counts for each gene from all three experiments (in cpm) were added and fractions belonging to each experiment were computed. Because the sum is constrained to be 100%, ternary plots could be used for plotting the three different fractions in two dimensions. For Fig 5E and 5F, for each gene shown we computed its percentile in correlation with DENV and ZIKV vRNA, i.e. the percentage of other host genes with a correlation less than this focal gene. This transformation emphasizes the top correlates/anticorrelates against batch effects and different multiplicities of infection in the DENV and ZIKV experiments. For Fig 5G and 5I, published tables of counts and metadata were downloaded from links present in each publication, normalized to counts per millions, and filtered for low-quality cells. We computed the correlation of host gene expression and vRNA in each experiment, then features with a high rank in at least one virus were selected and correlation coefficients were centered and normalized between -1 and 1 for each virus to enable meaningful cross-experiment comparison. Principal Component Analysis (PCA), Uniform Manifold Approximation and Projection (UMAP), similarity graphs, and Leiden clustering [39] were computed and plotted.

## Cell selection and normalization

The criteria to select cells were as follows: total reads > 300,000, gene counts > 500 and a ratio of ERCC spike-in RNA to total reads ratio < 0.05. Based on these criteria, 2004 out of 2301 cells were selected for downstream analysis. Due to the high viral copies of VEEV in cells infected for 12 and 24 hrs (more than 10%), traditional normalization (dividing by total reads) caused a bias which underestimated the expression of host genes. To avoid this, we normalized gene counts to ERCC total reads, since these are not affected by the virus. Each gene count column (including virus reads) was thus divided by ERCC total reads and then log transformed.

## Loss-of-function assays

siRNAs (1 pmol) were transfected into cells using Lipofectamine RNAiMAX transfection reagent (Invitrogen) 72–96 hours prior to infection with VEEV-TC-83-nLuc, non-reporter VEEV-TC-83, or wild type VEEV-TrD. Custom Cherry-Pick ON-TARGET plus siRNA library against 11 genes was purchased from Dharmacon (see S2 Table for gene and siRNA sequence details).

## Gain-of-function assays

Individual plasmids encoding 11 human genes or empty control vector were transfected individually into U-87 MG cells with Lipofectamine 3000 (Invitrogen) 48 hours prior to infection with VEEV-TC-83-nLuc.

## Viability assays

Viability was measured using alamarBlue reagent (Invitrogen) according to the manufacturer's protocol. Fluorescence was detected at 560 nm on an Infinite M1000 plate reader (Tecan).

## RNA extraction and qRT-PCR

Total RNA was isolated from cells using RNeasy Mini Kit (Qiagen). For host genes, reverse transcription mixtures contained 1 μg or 10 μl RNA and High-Capacity RNA-to-cDNA reverse transcription kit (Applied Biosystems). qRT-PCR mixtures were assembled using 50 ng or 5 μl cDNA and PowerUp SYBR Green Master Mix (Applied Biosystems). For VEEV

detection, qRT-PCR mixtures were assembled using 50 ng or 5 μl total RNA and QuantiTect Probe RT-PCR Kit (Qiagen). Amplification and analysis were performed using QuantStudio3 system (ThermoFisher Scientific). Primer sequences are listed in S3 Table.

### Western blot analysis

Cells were lysed in M-PER Mammalian Protein Extraction Reagent (Thermo Fisher Scientific). Protein lysates were run on 4%–12% Bis-Tris gels (Invitrogen), transferred onto PVDF membranes (Bio-Rad). Blots were blocked and blotted with anti-FLAGlag (Sigma-Aldrich, catalog F1804) antibody. Signal was detected using anti-mouse HRP-conjugated secondary antibody (Cell Signaling Technology, catalog 7076).

### Statistics

All statistical analysis were performed with GraphPad Prism software. *P* values or q values were calculated by 1-way ANOVA with either Dunnett's or false discovery rate (FDR) corrected multiple comparisons tests, respectively, as specified in each figure legend.

## Results

### ViscRNA-Seq reveals cell-to-cell heterogeneity in VEEV-TC-83 and host gene expression

To characterize the relation between viral and host cell transcriptional dynamics over the course of VEEV infection, human astrocytoma cells (U-87 MG) [40] were infected with VEEV-TC-83 (attenuated vaccine strain) capsid fusion reporter virus expressing GFP [28] at MOIs of 0.1 and 1 or mock infected, and harvested at six time points: 0.5, 1.5, 4, 6, 12, and 24 hpi (**Fig 1A**). Single cells were then isolated and processed by viscRNA-Seq, as described previously [23]. Since the VEEV RNA is polyadenylated, it can be captured by the standard poly-T oligonucleotide that hybridizes with host transcripts. Nevertheless, to improve vRNA capture and ensure coverage at the 5' end of the viral genome, two specific viral capture oligonucleotides, at positions 352 and 1,742 of the VEEV genome, were added to the reaction (see Methods). In total, 4608 cells were processed, of which 2301 cells were sequenced with approximately 1 million reads/cell (**S1A Fig**). 2004 cells passed quality controls and were analyzed (see Methods).

To identify a proper cutoff for defining infected cells, we analyzed both GFP signal and vRNA reads. During cell sorting (the first step of viscRNA-Seq) the GFP signal was recorded using the fluorescein isothicyanate (FITC) gate, enabling measurement of cellular GFP expression levels. The GFP signal was comparable in cells harboring 1 to 1000 viral reads, yet it sharply increased in cells harboring over 1000 viral reads (**Fig 1B**). The lower sensitivity of GFP signal relative to viral reads is likely due to the lag of protein expression after RNA synthesis and indicates that virus reads can be used as an effective indicator for VEEV infection. Next, we sought to define a cutoff to distinguish infected from bystander cells (uninfected but derived from the sample that was exposed to the virus). We set multiple cutoffs between 1 and 100 viral reads, selected only cells with viral read number greater than these cutoffs, and calculated the correlation coefficient between GFP expression and viral reads (**S1B Fig**). The correlation between GFP expression and viral reads first increased with the cutoffs and then stabilized once the cutoff reached 10 viral reads, with correlation coefficients greater than 0.8 via both Spearman's and Pearson correlations. We therefore defined the presence of 10 or more viral reads as the cutoff to distinguish VEEV-infected from bystander cells. Similar findings were observed upon plotting the relationship between GFP expression and virus/total

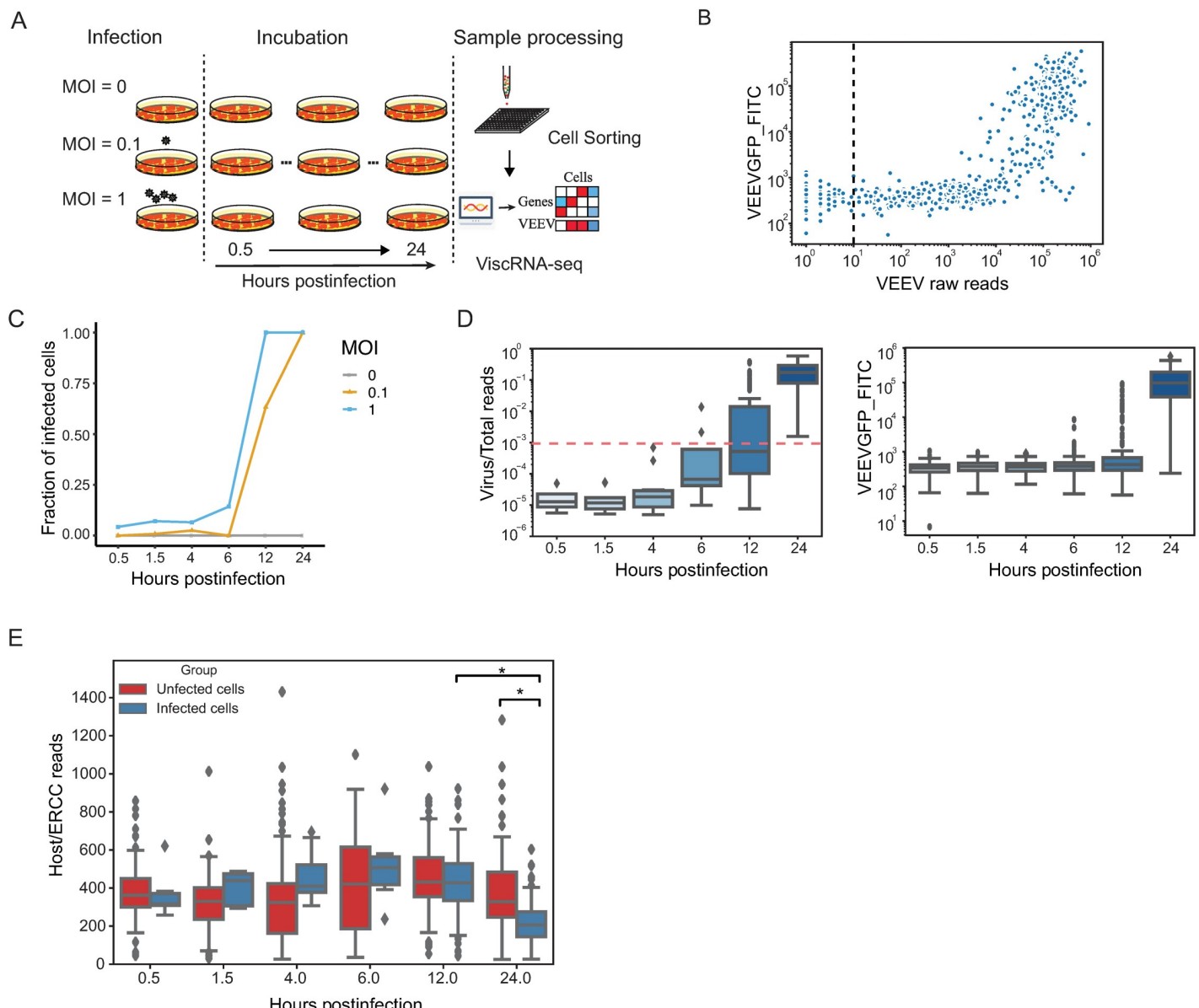

**Fig 1. Cell-to-cell heterogeneity during VEEV infection.** (A) Schematic of the experimental setup. (B) A scatter plot showing VEEV cDNA sequencing reads and GFP expression measured via FACS (FITC gate) in cells harboring 1 or more viral reads. The dotted line represents the cutoff of infected cells. Cells harboring more than 10 viral reads are considered infected. (C) The fraction of VEEV-TC-83-infected U-87 MG cells over time for two MOIs. (D) Box plots depicting the ratio of virus to total cDNA reads (left) and GFP expression level (right) over time. The horizontal dotted line represents the threshold dividing cells into "low vRNA" and "high vRNA" harboring cells (see text). (E) Box plots showing host cDNA to ERCC read ratio in infected and uninfected cells derived from different time points postinfection. *p < 0.05 by Mann-Whitney U test. MOI, multiplicity of infection; ERCC, External RNA Controls Consortium.

reads ratio (vs. raw viral reads) (**S1C Fig**), indicating that the selected threshold of 10 viral reads (or 0.00001 virus/total reads) is not affected by differences in sequencing depth between cells.

The fraction of VEEV-infected cells increased with both time and MOI and saturated at 12 and 24 hpi with MOI 1 and 0.1, respectively (**Fig 1C**). The infection status in these cells at the various time points postinfection was confirmed via qRT-PCR (**S1D Fig**). A rapid increase in the ratio of both viral/total reads and GFP expression was observed within single cells over

time (Fig 1D). Notably, the distributions of virus/total reads and GFP expression were particularly wide at 12 hpi when analyzing either the entire infected cell population or infected cells separated by the two MOIs (S1E and S1F Fig). At 24 hpi, the observed increase in vRNA reads was associated with a decline in cellular transcripts. The normalized cellular mRNA reads (calculated by dividing the absolute number of reads by the sum of External RNA Controls Consortium (ERCC) spike-in reads) declined in the infected cell group at 24 hpi relative to the corresponding uninfected cell group and the same infected cell group at 12 hpi (Fig 1E). To avoid an artificial decline in host gene reads in cells with high vRNA abundance, rather than normalizing cellular gene reads by the total reads, we normalized by ERCC reads for most downstream analyses. This transformation is akin to an estimate of the actual number of mRNA molecules for each gene (up to a constant factor).

## Altered expression of cellular factors and pathways during VEEV infection

The wide distributions of virus/total reads observed at 12 hpi suggested that to more precisely characterize the phenotype of cells from VEEV-infected samples, cells should be divided based on the virus/total read content rather than time postinfection or MOI. To identify host genes whose expression is altered during VEEV infection, we integrated differential gene expression and correlation analyses. First, we combined cells harvested at different time points. Since the GFP signal started to increase significantly with a virus/total read ratio greater than 0.001 (S1C Fig), we divided cells into the following three groups based on this cutoff: infected cells with high vRNA (virus/total reads > 0.001), infected cells with low vRNA (virus/total reads < 0.001), and uninfected controls (S2A Fig). Since GFP expression and viral reads correlated well in the high vRNA group, we focused on differences between the high vRNA cell group and the uninfected group. Computing differential expression at the distribution level (Mann-Whitney U test) revealed 1734 host genes, whose expression level significantly differed between the high vRNA group and the uninfected group. To test the robustness of the population division, we applied a set of cutoffs (ranging from 0.0001 virus/total reads to 0.01 virus/total reads) and computed differential expression between the high vRNA group and the uninfected controls based on each of these cutoffs (S2B Fig). The number of differentially expressed genes (DEGs) identified increased up to a cutoff of 0.001 virus/total reads and then plateaued. Moreover, DEGs identified by a cutoff of 0.001 largely overlapped (over 90%) with those detected with higher cutoffs, confirming that the cutoff of 0.001 is robust in distinguishing between infected cells with high and low vRNA abundance. We predicted that differential expression of some genes might be related to time effect resulting from differences in incubation duration rather than from viral infection. To control for such confounders, we calculated Spearman's correlation coefficients between gene expression and time postinfection. Genes whose expression was similarly altered over time between infected and uninfected cells were thought to represent time effect. 1707 of the 1734 DEGs between the high vRNA and uninfected groups passed this additional filter (Fig 2A).

In parallel, we computed Spearman's rank correlation coefficients between gene expression and vRNA abundance across all cells, as done previously for flaviviruses [23]. Our data indicate that the majority of host genes are negatively correlated with vRNA abundance (S2C Fig). Stratifying host genes by expression level in uninfected cells indicated that highly expressed genes demonstrated a stronger negative correlation with vRNA abundance (S2D Fig), suggesting that cellular functions relying on highly expressed genes are more vulnerable to VEEV infection. To identify genes that are both differentially expressed between infected and uninfected cells and correlated with vRNA, we computed the intersection between the 1707 DEGs with the top 600 genes that either positively (n = 300) or negatively (n = 300) correlated with vRNA. 263 overlapping genes emerged from this analysis (Fig 2A).

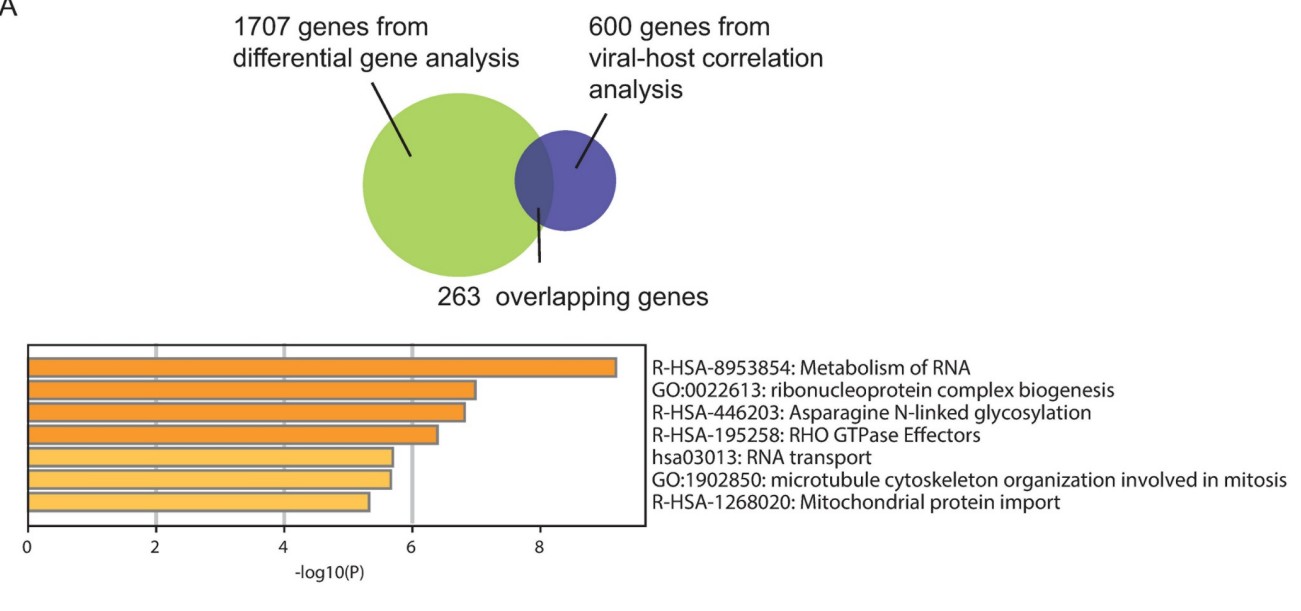

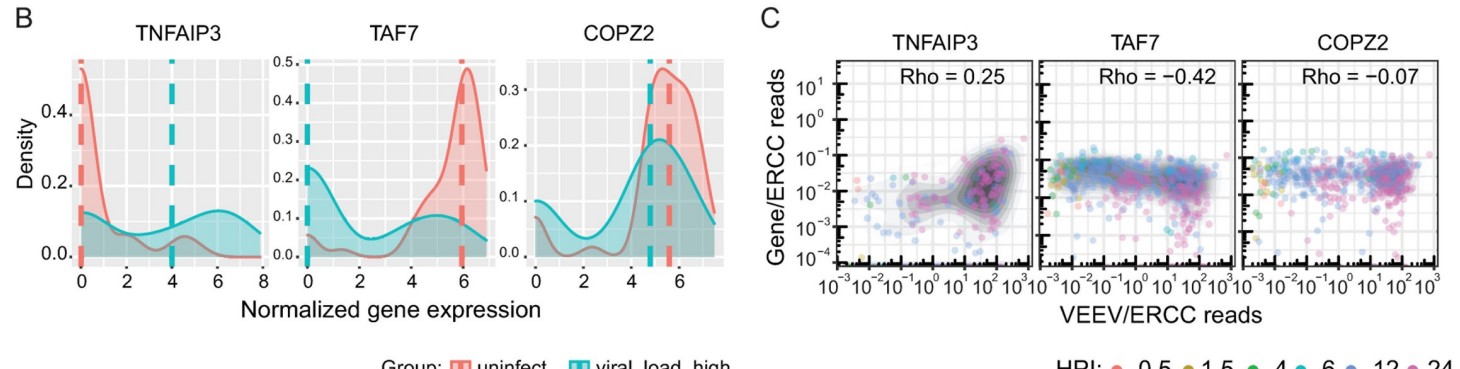

**Fig 2. Host genes and pathways are altered during VEEV infection.** (A) A Venn diagram showing the number of unique and overlapping genes that emerged from the differentially gene expression analysis and host RNV/vRNA correlation analysis. Molecular function terms and P values derived from Gene Ontology (GO) enrichment analysis of 263 genes that are both differentially expressed between high vRNA and uninfected cells and correlated with vRNA. (B) Ridge plots of representative host genes that are differentially expressed between high vRNA and uninfected cells and a gene (COPZ2) whose level is unaltered. 50 cells from each group were selected for plotting. Dash lines indicate median expression level of the corresponding genes. Gene expression was normalized using the following formula: ln ((gene counts / ERCC counts) + 1). (C) Representative scatter plots of host gene expression versus vRNA abundance and corresponding Rho Spearman's correlation coefficients. Each dot is a single cell colored by the time postinfection, and the shaded contours indicate cell density (greyscale, darker is higher). HPI, hours postinfection; MOI, multiplicity of infection; ERCC, External RNA Controls Consortium; TNFAIP3, Tumor Necrosis Factor Alpha-Induced Protein 3; TAF7, TATA-Box Binding Protein Associated Factor 7; COPZ2, COPI Coat Complex Subunit Zeta 2.

Gene Ontology (GO) enrichment analysis of these 263 genes via metascape [41] highlighted metabolism of RNA as the most enriched molecular function term (**Fig 2A**). Shown in **Figs 2B and 2C** and **S2E** are representative genes that were overexpressed in high vRNA cells vs. uninfected and low vRNA and positively correlated with vRNA (TNFAIP3), underexpressed and negatively correlated with vRNA (TAF7), or not differentially expressed and were uncorrelated with vRNA (COPZ2). The expression level of these genes did not change over time in uninfected cells, supporting that their altered levels represent actual differences between the groups rather than a time effect (**S2F Fig**).

## Early infected "superproducer" cells show distinct patterns of host gene expression

During cell processing, we noticed that 2% of the cells infected with an MOI of 1 at 6 hpi, the duration of a single cycle of VEEV replication [7,42], showed stronger GFP signals (FITC gate readout > 1000) than the remaining cells in the same condition. To probe the relevance of this unexpected finding, we specifically sorted these cells. In correlation with their GFP expression, the majority of these cells harbored ~100-fold higher virus/total reads ratio than the remaining cells in the same condition, suggesting that once initiated, viral replication proceeded extremely fast in these "superproducer" cells (**Fig 3A**). 11 cells were defined as "superproducer" cells based on the following criteria: harboring > 0.001 vRNA/total reads and GFP readout > 1000 at 6 hpi (MOI = 1) (**Fig 3A**). To elucidate whether these "superproducer" cells exhibit a distinct gene expression pattern, we conducted differential gene expression analysis (Mann-Whitney U test) between these 11 cells and uninfected cells as well as low vRNA harboring cells, both harvested at the same time point (6 hpi). A total of 16 DEGs were identified showing a distinct expression pattern only in these "superproducers", with representative overexpressed and underexpressed genes shown in **Fig 3B** and **3C**. Notably, these genes were also differentially expressed between the "superproducer" cells and high vRNA cells harvested at 24 hpi, suggesting that they do not represent a general response to high vRNA abundance, but rather a unique feature of this cell population. Among the overexpressed genes were SYTL3, a protein that directly binds to Rab27A to regulate Rab27-dependent membrane trafficking; KDM3B, a lysine demethylase; SNX29, a member of the sorting nexin family; and COG5, a component of Golgi-localized complex that is essential for Golgi function. Among the underexpressed genes were ZMAT5, an RNA-binding protein belonging to the CCCH zinc finger family of proteins implicated in antiviral immune regulation [43]; VPS37A, a component of the ESCRT-I protein complex; and AC087343.1, a ribosomal protein L21 pseudogene. These findings provide evidence that a small subset of "superproducer" cells largely drives VEEV replication during the first viral life cycle and demonstrates a distinct gene expression pattern. These results also point to SYTL3, KDM3B, SNX29 and COG5 as candidate proviral factors, and to ZMAT5, VPS37A and AC087343.1 as potential antiviral factors.

## The expression of genes involved in intracellular membrane trafficking correlates with the ratio of 3' to 5' vRNA reads

By including both a poly-T and a 5'-end specific capture oligonucleotides in the viscRNA-Seq, good read coverage at both ends of the VEEV genome was obtained (**Fig 4A**). We defined 5' RNA reads as those corresponding to the first 1,700 bases (encoding nonstructural proteins), and thus derived from the genomic vRNA only, and 3' RNA reads as those corresponding to the last third of the genome (encoding structural proteins), derived from both the genomic and subgenomic vRNAs (**Fig 4B**). The stoichiometry of the 3' and 5' RNAs was highly heterogeneous between cells. While at early stages of infection, the 3' to 5' (structural to nonstructural) vRNA read ratio (3'/5' read ratio), as defined by these genomic regions, was below or around 1, at late stages, it reached up to 4 and was correlated with total vRNA abundance (**Fig 4C**). In contrast, the read ratio between two segments we selected as internal controls at the 5' end of the vRNA (5'a/5'b read ratio) and between two segments at the 3' end (3'a/3'b read ratio) did not correlate with the cellular vRNA abundance (**Fig 4D and 4E**). To test the hypothesis that differences in vRNA stoichiometry are associated with distinct host responses, we measured the Spearman's correlation coefficients of all host genes with the 3'/5' read ratio in the same cell. The resulting histogram distribution curve revealed a tail of host genes whose expression increased with the 3'/5' read ratio (**Fig 4F**), in contrast to the distribution of host

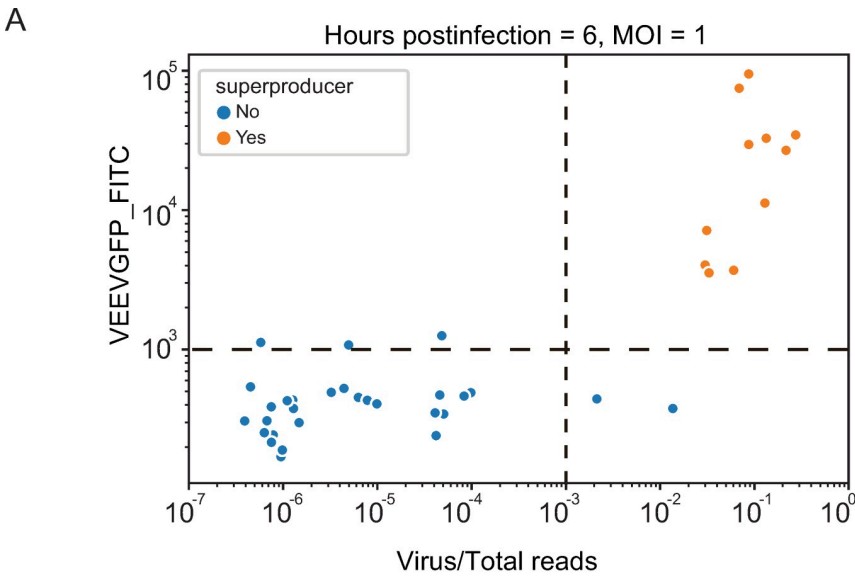

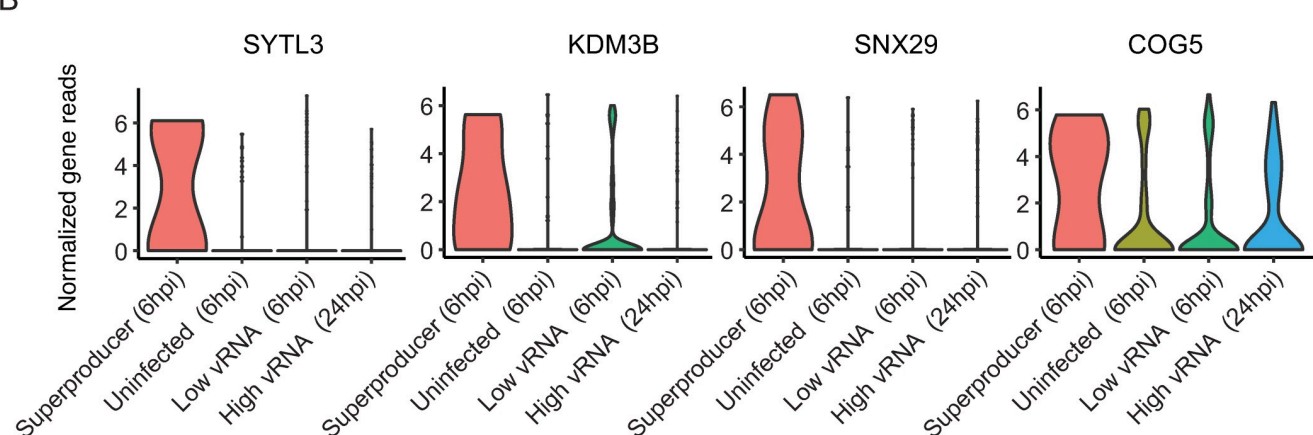

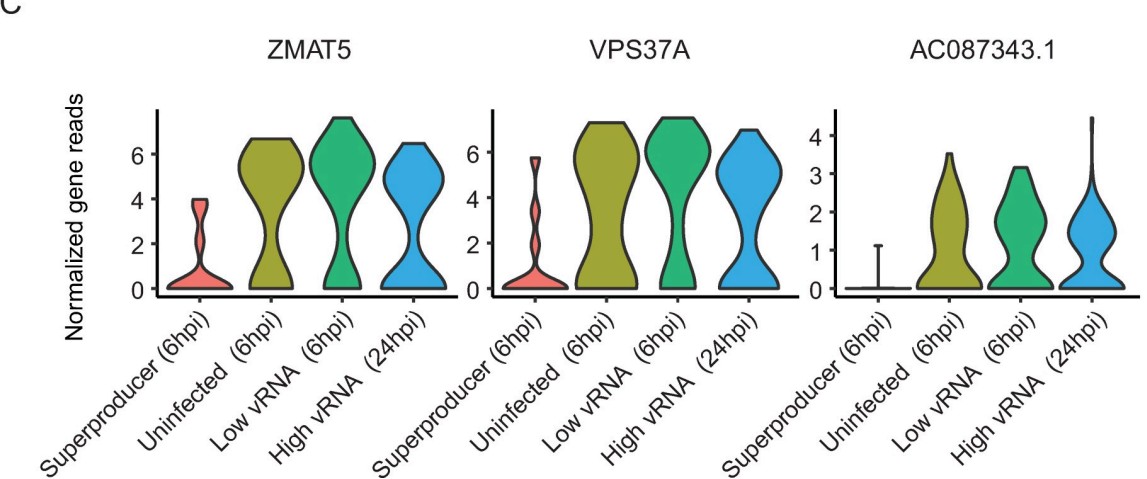

**Fig 3. "Superproducer" cells exhibit altered gene expression patterns.** (A) Scatter plot depicting GFP expression level and virus/total reads in cells at 6 hours postinfection (hpi) with VEEV- TC-83 at an MOI of 1. The horizontal and vertical dash lines indicate the cutoffs of GFP signal and virus/total read

ratio, respectively (see text). Each dot represents a cell. Orange, cells with a GFP signal readout that is greater than 1000 and virus/total read ratio greater than 0.001 defined as "superproducers" (n = 11); blue, cells not meeting these criteria. (B and C) Representative violin plots showing genes that are upregulated (B) or downregulated (C) specifically in "superproducer" cells relative to either uninfected cells, low vRNA cells harvested at 6 hpi or high vRNA cells harvested at 24 hpi. HPI, hours postinfection; MOI, multiplicity of infection. SYTL3, Synaptotagmin Like 3; KDM3B, Lysine Demethylase 3B; SNX29, Sorting Nexin 29; COG5, Component Of Oligomeric Golgi Complex 5; ZMAT5, Zinc Finger Matrin-Type 5; VPS37A, Vacuolar Protein Sorting-Associated Protein 37A; AC087343.1, Ribosomal Protein L21 (RPl21) Pseudogene.

genes in correlation with the total vRNA reads (S2C Fig). Positively correlated genes were mostly involved in various aspects of intracellular trafficking and included factors previously reported to be required for VEEV infection via an siRNA screen including ARP3 [19], RAC2, a paralog of RAC1 (19), and DDX5, a member of the DEAD box family of RNA helicases [20]. Novel factors among the positively correlated genes included factors involved in late endosomal trafficking (RAB7A [44], the accessory ESCRT factor (BROX) [45], and the SNARE protein VAMP7 [46]), as well as in ER to Golgi trafficking (SEC22B [47], regulation of secretion (PIP4K2A) [48], lysosome function and autophagy (LAMP2) [49], actin polymerization (PFN2) [50], and acidification of intracellular organelles for protein sorting (ATP6V1B2) [51] (Fig 4G). Accordingly, pathway analysis on the top 300 correlated genes identified macroautophagy, exocytosis regulation, membrane trafficking and vesicle organization as the highly enriched functions (Fig 4H). Notably, these genes were only positively correlated with the 3'/5' read vRNA ratio and not with the total vRNA reads. These findings indicate that the late stages of VEEV infection are characterized by heterogeneous stoichiometry of structural (3') and nonstructural (5') vRNAs and upregulation of intracellular trafficking pathways previously implicated in assembly and egress of various RNA viruses in cells with an excess of structural vRNA. Moreover, these results highlight the unique opportunity to discover candidate proviral factors for VEEV-TC-83 infection by correlating gene expression with specific viral genome stoichiometry via viscRNA-Seq.

In addition to enabling quantification of the 5' and 3' vRNA reads, the high coverage of the viral genome provided by viscRNA-Seq revealed rare structural viral read variants. The most common among these variants was a 36-base gap within the coding region of the 6K protein, whose presence was predicted to form a stable hairpin structure (S1 Text and S3 Fig). The biological relevance of this gap remains to be elucidated, and we cannot currently exclude that it could be a result of polymerase errors during library preparation. However, stable RNA structures play essential roles in viral replication and tropism across multiple viruses. Moreover, it is possible that this finding represents the presence of defective virus genomes, which have been observed in various RNA viruses [52,53].

## Validation of candidate proviral and antiviral factors

Next, we probed the functional relevance of 11 genes that either strongly or moderately correlated with vRNA abundance for viral infection. We first conducted loss-of-function screens by measuring the effect of siRNA-mediated depletion of these 11 individual genes on VEEV-TC-83 infection and cellular viability in U-87 MG cells (Fig 5A). The knockdown efficiency of the relevant transcripts was confirmed by qRT-PCR (S4A Fig). Depletion of CXCL3, ATF3, TNFAIP3, and CXCL2, four out of five genes tested that positively correlated with vRNA abundance via viscRNA-Seq (orange bars), reduced VEEV-TC-83 infection by more than 40% as measured by luciferase assay 18 hpi with a nano-luciferase reporter TC-83 virus (VEEV-TC-83-nLuc) and normalized to cellular viability in two independent screens, suggesting that they are candidate proviral factors (Figs 5A, S4B and S4C). In contrast, depletion of 3 of 6 genes tested that negatively correlated with vRNA (grey bars) enhanced VEEV-TC-83 infection, suggesting that these proteins may function as antiviral factors (Figs 5A, S4B and

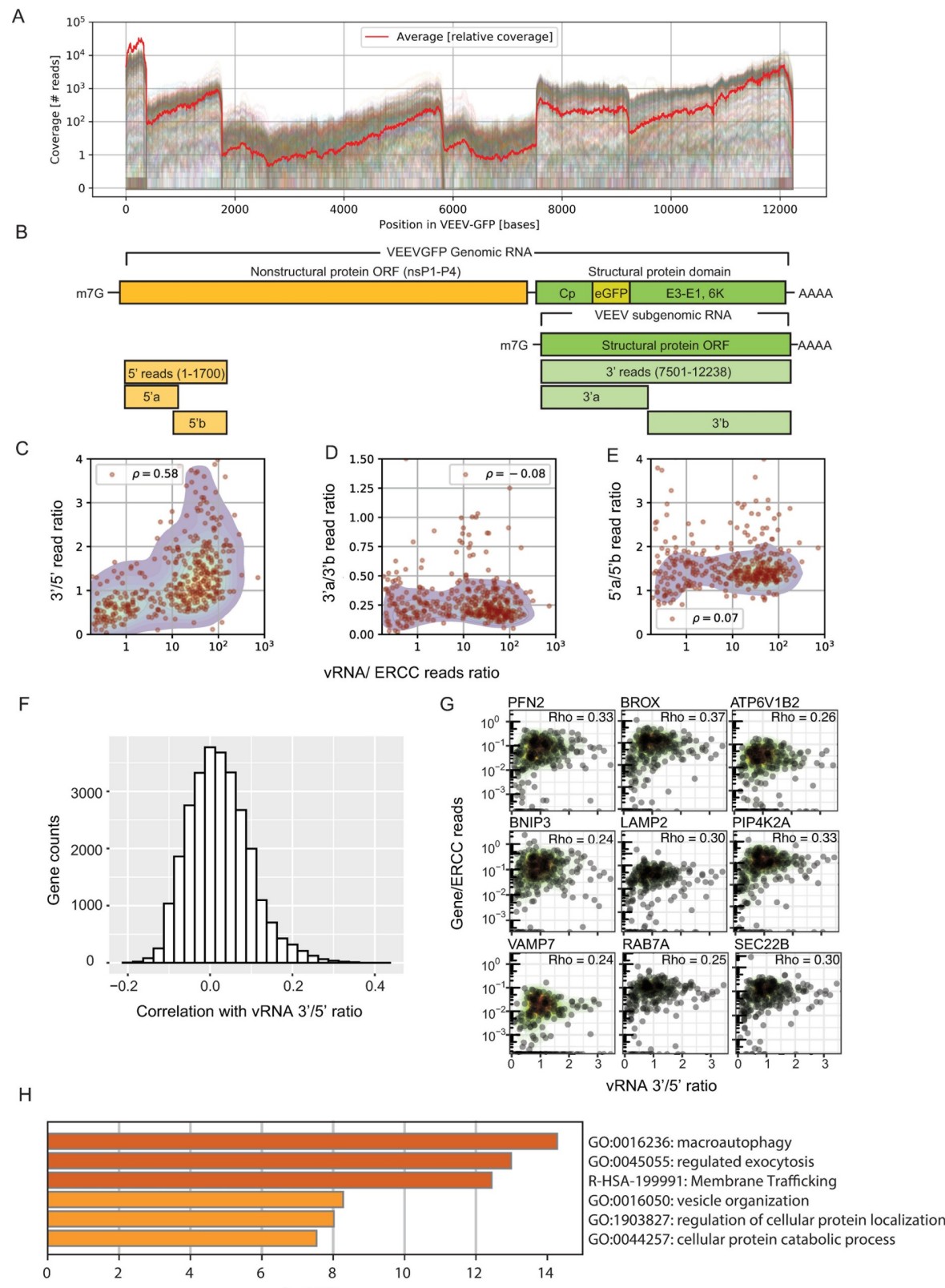

**Fig 4. The expression of genes involved in intracellular membrane trafficking correlates with the ratio of 3' to 5' vRNA reads.** (A) Coverage of viral reads over the entire VEEV-TC-83 genome. Each line is a cell, and the red line is a scaled average across all cells. (B)

Genome architecture of VEEV highlighting the nonstructural (yellow) and structural (green) protein domains. (C) Scatter plot showing positive correlation of VEEV 3'/5' read ratio with cellular vRNA abundance. Each dot is an infected cell. (D-E) Scatter plots showing no correlation between the 3'a/3'b read ratio (D) and 5'a/5'b read ratio (E) and cellular vRNA abundance. (F) Histogram of Spearman's correlation coefficients between all host genes and the 3'/5' read ratio. (G) Representative scatter plots of host gene expression versus vRNA 3'/5' read ratio and corresponding Rho Spearman's correlation coefficients. Each dot is a cell and contour plots indicate cell density (low to high, green to red). (H) Gene enrichment analysis of top 300 genes positively correlated with the 3'/5' read ratio. ORF, opening reading frame; PFN2, Profilin 2; BROX, BRO1 Domain- And CAAX Motif-Containing Protein; ATP6V1B2, ATPase H+ Transporting V1 Subunit B2; BNIP3, BCL2 Interacting Protein 3; LAMP2, Lysosomal Associated Membrane Protein 2; PIP4K2A, Phosphatidylinositol-5-Phosphate 4-Kinase Type 2 Alpha; VAMP7, Vesicle Associated Membrane Protein 7; RAB7A, Ras-Related Protein Rab-7a; SEC22B, SEC22 Homolog B.

S4C). Suppression of PPP2CA demonstrated no effect on viral infection, suggesting that it is either non-essential or not restricting (possibly due to redundancy in host factor requirement) (**Figs 5A**, **S4B** and **S4C**).

In parallel, we conducted gain-of-function screens by ectopically expressing the same 11 individual gene products in U-87 MG cells followed by VEEV-TC-83-nLuc infection (**Fig 5B**). The level of ectopic expression of these factors was confirmed by Western blotting (**S4D Fig**). Using a cutoff of greater than 40% change in viral infection normalized to cell viability in two independent screens, overexpression of most genes resulted in an inverse effect to that observed with the siRNA, i.e. if knockdown inhibited viral infection, overexpression enhanced it and vice versa (**Figs 5A, 5B**, **S4E** and **S4F**). Overexpression of CXCL2, TNFAIP3, ATF3, and CXCL3 increased VEEV-TC-83 infection, suggesting rate limitation associated with these candidate proviral factors (**Figs 5B**, **S4E** and **S4F**). In contrast, overexpression of the majority of the anticorrelated gene products reduced VEEV-TC-83-nLuc infection via luciferase assays, suggestive of an antiviral phenotype (**Figs 5B**, **S4E** and **S4F**).

To further probe these findings, we conducted loss-of-function experiments in U-87 MG cells infected with non-reporter VEEV-TC-83. Viral titers were measured via plaque assays in culture supernatants and intracellular vRNA levels were measured via qRT-PCR assays in lysates derived from cells transfected with individual siRNAs targeting 10 of the 11 cellular factors or with non-targeting (NT) control siRNA 24 hours postinfection (PPP2CA was excluded since altering its expression level did not impact infection). Similar to the luciferase assay results, suppression of CXCL2, TNFAIP3, ATF3, and CXCL3 expression reduced both the viral titer and intracellular vRNA (**S5A and S5B Fig**) with no effect on cell viability (**S5C Fig**), supporting a proviral phenotype. Depletion of TAF7, SURF4, and RAB1A, whose transcript levels anticorrelated with vRNA abundance, increased the luciferase signal (**Fig 5A**) but decreased the infectious viral titers and vRNAs in cells infected with the non-reporter VEEV-TC-83 (**S5A and S5B Fig**), highlighting differences between these assays, which measure different aspects of the viral life cycle, and suggesting a possible proviral role. Moreover, while the transcriptional level of TRMT10C and EIF4A3 anticorrelated with vRNA abundance, their gene products demonstrated a proviral phenotype in most assays (**Figs 5A, 5B**, **S5A** and **S5B**). These findings highlight the challenge of identifying antiviral factors within the context of a host transcriptional shutdown. The discrepancies observed with antiviral candidates may result from regulation of these genes at the translational level or from downstream effects of these multifunctional genes.

To probe the relevance of these findings in virulent VEEV infection, we measured the effect of depletion of these cellular factors via siRNAs on VEEV-TrD (Trinidad Donkey) strain infection via plaque assays at 24 hpi. Depletion of CXCL3 and EIF4A3 dramatically reduced VEEV-TrD titers (**S5D Fig**), similarly to the effect seen with TC-83 (**S5B Fig**). Depletion of TNFAIP3, ATF3, TAF7, and TRMT10C mildly to moderately reduced VEEV-TrD infection (**S5D Fig**). Interestingly, depletion of SURF4, an ER cargo receptor, increased virulent

A

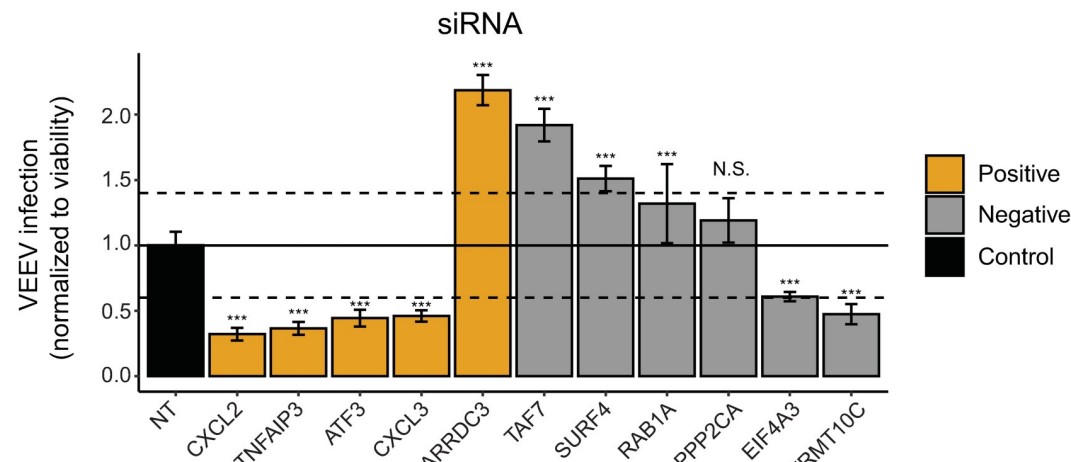

B

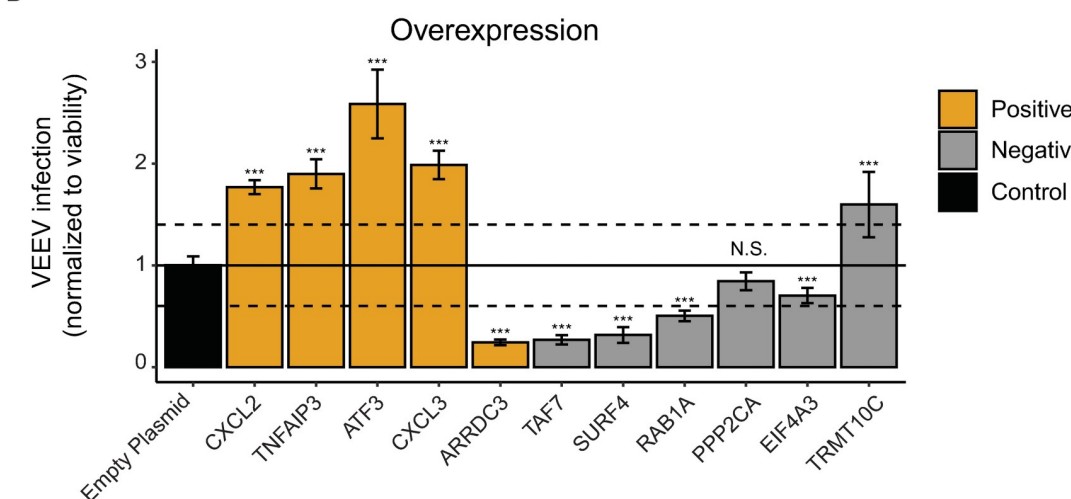

C

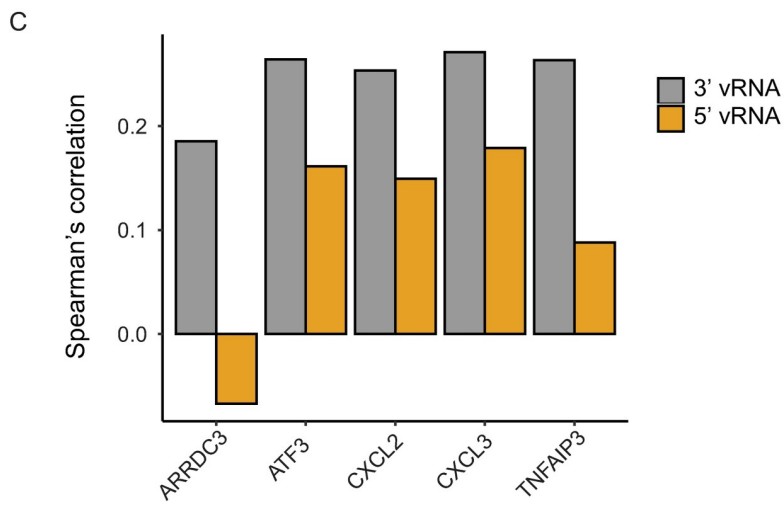

**Fig 5. Validation of candidate VEEV-TC-83 proviral and antiviral genes.** VEEV-TC-83 infection relative to non-targeting (NT) siRNA (A) or empty plasmid (B) controls following siRNA-mediated knockdown (A) or overexpression (B) of the indicated host factors measured by luciferase assays at 18 hpi (MOI = 0.01) of U-87 MG cells with VEEV-TC-83-nLuc and normalized to cell viability. Columns are color-coded based on the correlation of the respective gene with vRNA abundance via viscRNA-Seq: orange for genes that are positively correlated with vRNA and grey for genes that are negatively correlated with vRNA. Both data sets are pooled from two independent experiments with six replicates each. Shown are means ± SD; *p < 0.05, **p < 0.01, ***p < 0.001 relative to the respective control by 1-way ANOVA followed by Dunnett's post hoc test. The dotted lines represent the cutoffs for positivity. Confirmation of altered level of expression and cellular viability measurements are shown in S4 Fig. (C) Correlation coefficients between proviral candidates with the 3' (grey) and 5' (orange) vRNA reads.

VEEV-TrD infection, suggesting an antiviral effect in agreement with the prediction based on the viscRNA-Seq data. However, siRNA knockdown of SURF4 decreased the viral RNA levels and the number of infectious particles released during infection with the attenuated VEEV-TC-83 strain (S5D Fig). In contrast, depletion of CXCL2 and RAB1A, which suppressed TC-83 viral titers (S5B Fig), had no apparent effect on TrD infection (S5D Fig).

ARRDC3, a member of the arrestin family [54], was positively correlated with vRNA abundance, yet its depletion increased VEEV-TC-83 infection via most assays as well as VEEV-TrD infection (Figs 5A, S5A and S5D), and its overexpression decreased infection (Fig 5B), in contrast with the other four positively correlated genes tested. To probe this discrepancy, we measured the correlation of ARRDC3 expression with the 5' and 3' vRNA reads separately. Notably, ARRDC3 reads positively correlated with the 3' vRNA reads but negatively correlated with the 5' vRNA reads. In contrast, the other four proviral candidates positively correlated with both the 5' and 3' vRNA reads (Fig 5C). This finding suggests that ARRDC3 might have a dual function during VEEV infection. Together, these findings highlight the utility of viscRNA-Seq in identifying candidate proviral and antiviral factors.

## Comparative viscRNA-Seq analysis across five RNA viruses reveals distinct and common cellular pathways affected by viral infection

To define the elements of the host response that are unique to VEEV or common across multiple unrelated viruses, we first compared the VEEV dataset with our previously published viscRNA-Seq data from human hepatoma (Huh7) cells infected with DENV and ZIKV [23]. Since the baseline gene expression levels in astrocytes (VEEV-TC-83) are different from those in hepatocytes (DENV, ZIKV), we limited the analysis to genes that were similarly expressed (within a 10-fold change) in uninfected Huh7 and U-87 MG cells. We selected cells with greater than 2 vRNA reads per million joint (viral + host) reads and monitored how the expression of host genes changes with increasing vRNA abundance across the three infections. In all three viral infections, the majority of host genes were not correlated with vRNA abundance. Nevertheless, a number of host genes exhibited correlations with one or more viruses. Three robust patterns were identified (Fig 6A–6C): genes, such as HSPA5, that were upregulated in DENV infection and downregulated in ZIKV and VEEV infections (Fig 6A); genes like NRBF2 that were upregulated only during ZIKV infection (Fig 6B); and genes, such as SERP1, that were downregulated only in VEEV infection (Fig 6C). No genes that were upregulated only in VEEV infection could be identified. Beyond these general categories, the resulting patterns of viral and host expression were, however, quite complex.

To circumvent the masking effect of VEEV transcriptional shutdown, we then compared the genes that positively correlated with the 3'/5' VEEV RNA ratio with those positively or negatively correlating with DENV or ZIKV vRNA (Fig 6D). This analysis revealed genes, such as BROX, GEM, and RNF114 that are positively correlated with the respective vRNA in all three viral infections, genes, such as CTSB and SPTLC1 that are positively correlated with 3'/5' VEEV RNA and ZIKV but not DENV vRNA, and genes that are positively correlated with 3'/

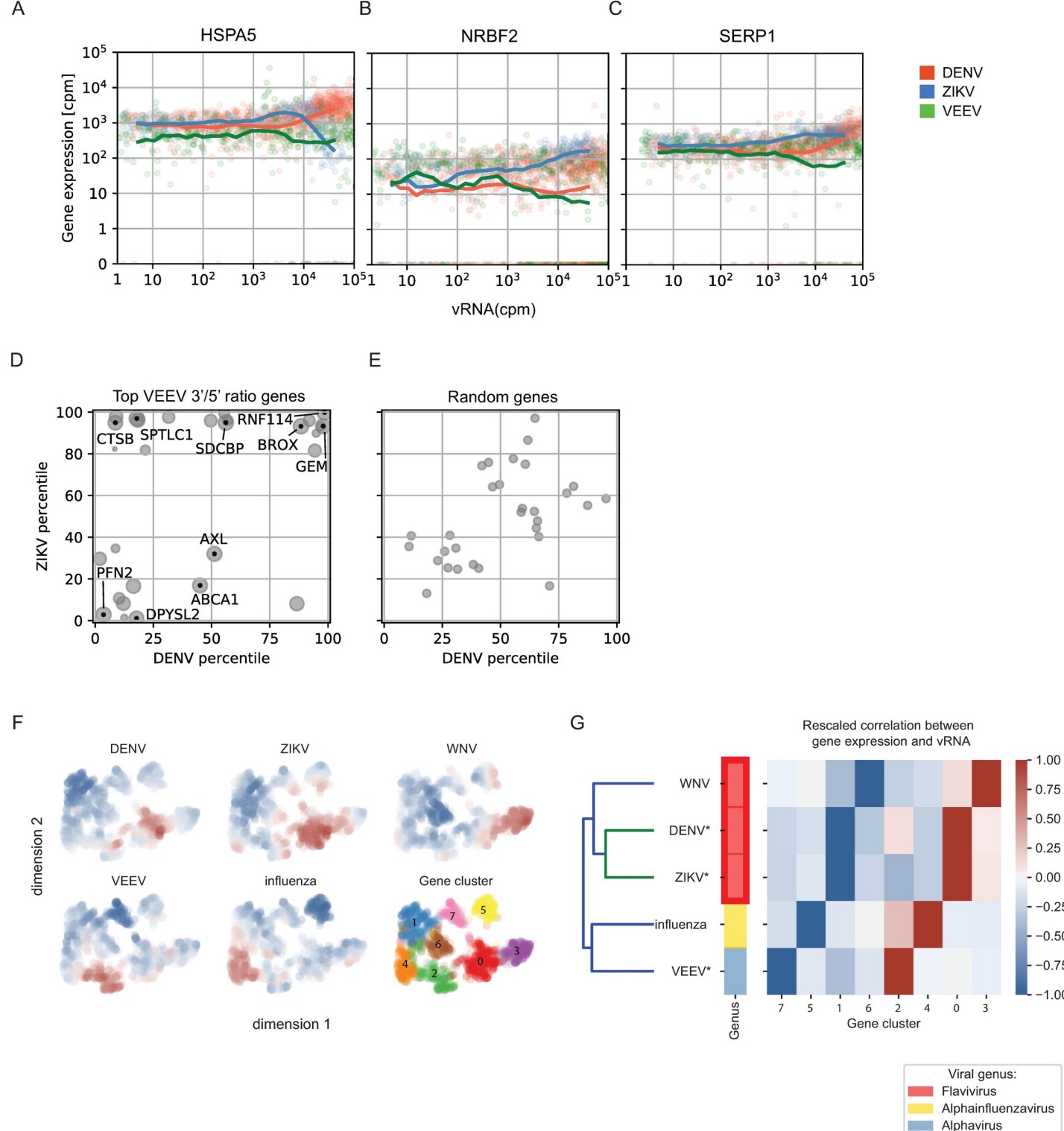

**Fig 6. Comparative viscRNA-Seq analysis across five RNA viruses.** (A-C) Scatter plots of representative host gene expression versus vRNA in single cells during DENV (orange), ZIKV (blue), and VEEV-TC-83 (green) infection. Dots indicate single cells, lines are running averages at increasing vRNA abundances. (D, E) Correlation between host gene expression and vRNA abundance during DENV versus ZIKV infection of the top genes that positively correlate with the VEEV 3'/5' read ratio (D) or a similar number of random genes (E). Each dot is a gene and the axis coordinate is the percentage of genes with a correlation with vRNA smaller than the the gene of interest. For (D), size of the dot increases with the correlation with VEEV 3'/5' read ratio (top correlated gene is the largest). (F) UMAP [56] embedding of

the correlations of host genes with vRNA during infection by 5 individual RNA viruses. Blue and red indicate downregulation and upregulation during each infection, respectively. Several clusters of genes are observed (0–7). (G) Hierarchical clustering of host gene clusters highlighting that gene upregulation is mostly virus-specific and is consistent with the known phylogeny. cpm, count per million; WNV, West Nile virus; IAV, influenza A virus.

5' VEEV RNA but negatively correlated with DENV and ZIKV vRNA, such as PFN2 and DPYSL2. In contrast, no significant correlations were observed when a comparable number of random genes were similarly analyzed (**Fig 6E**). Pathway analysis on genes that are positively correlated with both the 3'/5' VEEV RNA ratio and the two flaviviral RNAs identified ER processing, glycosylation, SELK (part of Endoplasmic-Reticulum-Associated Degradation), tRNA synthesis, protein folding, virion assembly, and intracellular transport as the highly enriched functions (**S6A Fig**). In contrast, cell cycle and apoptosis regulation were the most highly enriched functions in pathway analysis on genes that were positively correlated with 3'/5' VEEV RNA ratio but negatively correlated with the two flaviviral RNAs (**S6B Fig**). These results provide evidence that complex temporal dynamics exist across infections with different RNA viruses and highlight both common and unique cellular pathways that are altered by VEEV and flaviviruses.

Next, we expanded our comparative analysis by including published datasets derived from single-cell transcriptomic studies on different cell lines infected with two additional RNA viruses, influenza A virus (IAV) [24] and Weste Nile virus (WNV) [26] generated via 10x Genomics and Smart-seq2, respectively. Because different cell lines were used for different viruses, we calculated the ranks of the correlation coefficients between the expression of each host gene and vRNA for each virus, restricted the selection to the top and bottom 200 genes, and normalized the results between -1 and 1 for each virus. We then calculated the network of similarities between genes [55]. Uniform Manifold Approximation and Projection (UMAP) for Dimension Reduction [56] and Leiden clustering [39] of the genes highlighted 8 gene clusters with different expression patterns during various viral infections (**Fig 6F**). To understand the meaning of these clusters, we performed double hierarchical clustering and observed that clusters 2, 4, 0, and 3 were upregulated, while clusters 7, 5, 1, and 6 were mostly downregulated during viral infection (**Fig 6G**). DENV and ZIKV shared clusters for both upregulation and downregulation, as expected from their evolutionary proximity. The dendrogram of the five viruses was qualitatively consistent with the known phylogeny as derived from viral genomic sequences, which could indicate ancestral phenotypic signatures.

Overall, our analysis indicates that although comparing single cell viral infection data across species, cell lines, and technologies still presents challenges, this approach is informative in highlighting host genes and pathways that are commonly affected across very different viral families.

## Discussion

We and others have recently characterized the cellular response in virus infected cell lines [23,24], primary cells [26,57] and patient samples [27] via single-cell RNA-seq approaches. Moreover, we reported unique and overlapping determinants in the host response to two related flaviviruses at a single cell resolution [23]. Nevertheless, the host transcriptomic response to infection with alphaviruses, which induces a profound transcriptional shutdown of host genes, has not been previously characterized at a single cell level, and the single-cell transcriptomic responses of unrelated viruses have not been compared. By applying viscRNA--Seq to study the temporal infection dynamics of VEEV-TC-83 in human astrocytoma cells, we revealed large cell-to-cell heterogeneity in VEEV and host gene expression, transcriptomic signatures in distinct cell subpopulations, and candidate proviral and antiviral factors, some of

which we then validated. Additionally, we established a role for viscRNA-Seq in comparative evolutionary virology by demonstrating structural variants within the VEEV genome as well as unique and overlapping host gene responses across multiple clades of RNA viruses. These findings provide insights into the virus-host determinants that regulate VEEV-TC-83 infection and highlight the utility of viscRNA-Seq approaches and comparative single-cell transcriptomics.

A prominent feature of VEEV infection is a profound suppression of cellular transcription [6]. Nevertheless, it remained unknown whether this transcriptional shutdown globally affects all host mRNAs. Computing the distributions of vRNA expression in correlation with 5 groups of genes, distinguished by the level of gene expression in uninfected cells, demonstrated that highly expressed genes are more likely to be negatively correlated with vRNA abundance than genes that are expressed at a lower level. The cellular energy and machinery required to maintain a high level of gene expression likely play a role in increasing the vulnerability of highly expressed cellular genes to VEEV-induced transcriptional shutdown.

We have previously reported the utility of viscRNA-Seq in discovering functional transcriptomic signatures and candidate pro- and antiviral factors of DENV and ZIKV infections [23,27]. Nevertheless, the high replication rate of VEEV and the transcriptional shutdown it induces challenged our ability to detect alterations in gene expression and identify pro- and antiviral factors. To overcome these challenges, we used several strategies. First, since the viscRNA-Seq analysis revealed large differences in vRNA abundance between cells infected with the same MOI and harvested at the same time point, we stratified cell populations based on vRNA abundance rather than time postinfection. Integrating differential gene expression and correlation analyses of vRNA abundance with gene expression across the entire human transcriptome facilitated the discovery of 263 genes that were both differentially expressed between the high and mock infected controls and correlated with total vRNA. siRNA-mediated depletion and overexpression of a subset of these genes revealed that overall, genes involved in cytokine production, plus ATF3, a transcription factor commonly expressed in response to cellular stress, and TNFAIP3, an inhibitor of NFκB signaling, demonstrated a phenotype consistent with a rate-limiting proviral function. ARRDC3, one of 5 genes that were both differentially expressed and positively correlated with total vRNA, demonstrated a phenotype consistent with antiviral rather than a proviral effect. Interestingly, when studied in correlation with the individual vRNA transcripts, ARRDC3, a signaling arrestin family protein and a cargo-specific endosomal adaptor, was positively correlated with the 3' vRNA but negatively correlated with the 5' vRNA, suggesting that it may have a proviral effect during later stages and an antiviral effect in earlier stages of replication. By capturing such complex dynamics and not relying on averaging signals at distinct time points postinfection for stratification, the viscRNA-Seq approach may have an advantage over bulk sample knockdown or knockout approaches in identifying factors required for or restrictive of VEEV infection.

Whereas the effect of altered expression level of factors predicted to be proviral based on viscRNA-Seq on TC-83 infection was largely consistent between the various functional assays, the phenotypes observed among the candidate antiviral factors whose transcript level anticorrelated with vRNA were less obvious. For example, depletion of TAF7, SURF4, and RAB1A increased the luciferase signal in VEEV-TC-83-nLuc infected cells but decreased the infectious viral titer and intracellular vRNA in cells infected with the non-reporter VEEV-TC-83 strain, supporting a possible proviral rather than antiviral role. It is possible that the reduction in intracellular vRNA measured upon depletion of these factors reduced the translational shutdown induced by nsP2 [58], thereby increasing the luciferase signal. Similarly, the gene products of TRMT10C and EIF4A3 demonstrated a proviral phenotype in most assays. This difference between proviral and antiviral candidates is due to the asymmetric effect on gene

expression caused by the host transcriptional shutdown. Since VEEV causes a global downregulation of the host transcriptome, it becomes challenging to distinguish *in silico* genes that are downregulated further because of specific virus-host interactions. In contrast, even weak positive correlations between vRNA abundance and host gene expression are suggestive of genes that are "spared" from the global transcriptional shutdown. Biologically, these findings may result from differential regulation of the antiviral candidates at the translational level or from downstream effects of these multifunctional genes. The phenotype exhibited by ARRDC3 in both TC-83 and TrD infections and by SURF4 in the context of TrD infection, were consistent with potential antiviral effects. Moreover, in the case of ARRDC3, the relative ratio of 5'/3' viral transcripts (see below) was more in line with infectivity experiments than total amount of intracellular vRNA indicating that virus-host interactions are not fully recapitulated by a single correlation coefficient.

The high resolution provided by viscRNA-Seq enabled us to further focus on distinct cell populations, which facilitated identification of additional transcriptomic signatures. We discovered a subpopulation of cells demonstrating unusually high viral replication upon completion of a single cycle of viral replication. Importantly, this cell subpopulation is associated with host cell gene expression that is distinct from cells harboring lower vRNA at the same time. It is intriguing to speculate that overexpression of the identified hits involved in intracellular membrane trafficking (such as SYTL3, SNX29 and COG5) concurrently with underexpression of factors implicated in antiviral immune responses (such as ZMAT5) in this cell population drive the rapid increase in viral replication during the first viral lifecycle.

To further increase the resolution of our analysis, we took advantage of the ability of viscRNA-Seq to detect the two VEEV-TC-83 transcripts. A prior study on IAV has detected different levels of various segments of the viral genome across cells and investigated how this finding relates to successful virion production [24]. Similarly, analysis of the stoichiometry of the 5' and 3' RNA reads of VEEV, a non-segmented virus, revealed a large cell-to-cell heterogeneity. Moreover, the 3'/5' vRNA ratio substantially increased at late stages of infection, consistent with previous reports with other alphaviruses [10]. Remarkably, the histogram distribution curve of the Spearman's correlation coefficients of all host genes with the 3'/5' read ratio in the same cell revealed a long tail of host genes whose expression increased with the 3'/5' read ratio. Our findings indicate that these changes in stoichiometry of the vRNA transcripts during late stages of VEEV infection are associated with upregulation of distinct genes, particularly those involved in intracellular trafficking pathways. Notably, detection of these factors was only possible by correlating their expression specifically with the 3'/5' vRNA ratio and not the total vRNA reads. The involvement of these factors specifically in cells harboring high 3'/5' vRNA read ratio thus makes it experimentally challenging to further study them via bulk sample approaches. Nevertheless, it is tempting to speculate that some of the discovered late endosomal trafficking and lysosomal proteins (RAB7A [44], BROX [45], VAMP7 [46] and LAMP2 [49]) may be involved in forming the CPV-I composed of modified endosomes and lysosomes in which VEEV RNA replication occurs [13–15,59–62], and that ATP6V1B2 [51] may mediate the acidification of this acidic intracellular compartment [42]. Moreover, the positive correlation of proteins involved in ER to Golgi trafficking (SEC22B) [47], regulation of secretion (PIP4K2A) [48], autophagy (LAMP2) [49], actin polymerization (PFN2) [50], and ESCRT machinery (BROX, a Bro1 domain-containing protein like ALIX) [45,63], TSG101 and STAM2) with the 3'/5' vRNA read ratio proposes roles for these factors in late stages of the VEEV lifecycle, such as trafficking of the CPV-IIs to the plasma membrane, virion assembly, and/or budding [16–18]. These results propose a model wherein specific genes are upregulated within the profound transcriptional downregulation in a stoichiometry-dependent manner, and further illuminate the utility of viscRNA-Seq in identifying candidate

proviral and antiviral factors, including druggable candidates for host-targeted antiviral approaches.

One limitation of our study is the utilization of the vaccine (TC-83), rather than wild type strain of VEEV in the viscRNA-seq experiments. TC-83 was developed by serial passaging of the virulent, subtype IAB TrD VEEV strain [64]. While TC-83 maintains some degree of virulence manifesting with systemic illness and high level viremia in ~40% of vaccinated people [65,66] and horses [67], and with significant morbidity and mortality in mice upon subcutaneous or intracerebral inoculation, respectively [68], it is attenuated relative to wild type VEEV. Its attenuated phenotype results from two point mutations; one in nucleotide G3A within the 5' UTR, which increases the sensitivity of the virus to IFN-beta treatment [69] and the other is a Thr-to-Lys substitution in residue 120 of the E2 protein, which increases viral binding to heparan sulfate on the cell surface and renders the virus less lethal in mice [70–72]. These properties make the TC-83 strain suboptimal for studying some aspects of the immune response to VEEV infection, particularly at the organism level. Another limitation of the study is the use of the U-87 MG cell line rather than primary human cells. U-87 MG is the most widely used cell line for investigating VEEV-host interaction, with some examples reported here [73,74]. However, since derived from malignant glioma, like many other transformed cell lines, U-87 MG cells have altered type 1 IFN responses [75]. Combined, these factors have restricted our ability to capture some elements of the authentic host response to wild type VEEV infection, such as type I interferon response, and might limit the relevance of some of the findings in the context of *in vivo* infection.

Nevertheless, the key advantage of using the U-87 MG cell line is that it enabled the detection of subtle expression changes that would have been harder to detect in interferon-competent cells. These cells typically exhibit a substantial upregulation of numerous interferon-stimulated and related genes, masking important pathways associated with viral replication (vs. innate immunity). Indeed, when comparing DENV infection in IFN-deficient Huh7 cells [23] with infection in primary immune cells [27], this was precisely the dominant difference we observed. Moreover, our functional experiments demonstrate that while some findings, such as the requirement for CXCL2 observed in TC-83 infection, are not relevant to virulent VEEV-TrD infection, the requirement for CXCL3, EIF4A3, ATF3 and TAF7, and the potential antiviral effect of ARRDC3 may be functionally relevant to infection with both viral strains. Lastly, this study establishes the feasibility and utility of applying viscRNA-Seq to study other viruses from the VEE complex. The comparison between TC-83, its parental strain TrD, and other subtypes within the VEE complex could give an insight into viral evolution and virus-host interactions. In addition, applying viscRNA-Seq to study other alphaviruses can illuminate distinct host responses caused by viruses belonging to different subgroups (encephalitic vs arthritogenic).

Comparative evolutionary virology is an ideal application for single cell technologies because of the degree of genomic and functional diversity of infections. As a proof of concept, we compared the effect of unrelated human RNA viruses on the host cell in permissive cell lines. To address the confounding effect of different cell line backgrounds, we restricted the analyses in **Fig 6A–6F** to genes with a similar baseline expression level across cell lines. We compared genes that positively correlated with the 3'/5' VEEV RNA ratio with those correlating with DENV or ZIKV vRNA and found concordant signal for genes involved in protein processing and transport, whereas some cell cycle and apoptosis genes appeared to be specific to VEEV. When comparing data on five different viruses derived using different cell lines and technologies, we observed that while the closely related flaviviruses DENV and ZIKV affect a highly overlapping set of genes in both up and downregulation, more distant evolutionary relationships between the viruses lead to essentially distinct lists of dysregulated host genes.

Moreover, the "correct" viral phylogeny grouping of all three flaviviruses as a monophyletic group could be recovered purely from the host transcriptome perturbations, i.e. without using viral genomic information, which is intriguing. More viruses across the viral phylogeny should be assessed to evaluate whether this signal is the result of conserved ancestral function or, alternatively, of convergent functional evolution.

Overall, our study uncovered global and gene-specific host transcriptional dynamics during VEEV-TC-83 infection in a human astrocytoma cell line at single cell resolution and presented a novel approach to elucidate the evolution of virus-host interactions.

## Supporting information

**S1 Text. Rare structural viral read variants correlate with expression of specific host genes.** (DOCX)

**S1 Fig. Quality control and definition of infected cells.** (A) Quality control of the VEEV-TC-83 infected cells dataset. Shown are the total number of reads (left panel), gene counts (middle panel) and ratio of ERCC spike-in RNA reads to total reads (right panel). Cell selection criteria included: total reads > 300,000, gene counts > 500 and a ratio of ERCC spike-in RNA to total reads < 0.05. (B) Spearman's and Pearson correlation coefficients between GFP expression and vRNA reads when using different cutoffs (from 0 to 100) to define infected cells. (C) Scatter plots showing GFP expression level and vRNA/total reads in cells with detectable vRNA reads. Cells harboring more than 10 vRNA reads. Orange, infected cells; blue, bystander cells. (D) Genome copies of VEEV-TC-83 per 500 ng total RNA at various time points postinfection at MOI 0.1 or 1. (E and F) Box plots showing virus/total read ratio (left panel) and GFP expression level (right panel) over time in infected cells (viral reads > 10) at an MOI of 1 (E) or 0.1 (F) (no infected cells were detected at an MOI of 0.1 at 0.5 and 6 hpi). HPI, hours postinfection; MOI, multiplicity of infection; ERCC, External RNA Controls Consortium. (TIF)

**S2 Fig. Subgrouping cells based on viral load, representative differentially expressed genes (DGEs) and correlation analysis.** (A) Percentage of low and high vRNA-harboring cells at each time point. High vRNA-harboring cells are defined as virus cDNA reads/total cDNA reads > 0.001. (B) The number of differentially expressed genes (left panel) and cells with high vRNA abundance (right panel) under different cutoffs set by a range of virus cDNA/total reads (from 0.0001 to 0.01). (C) Distribution of Spearman's correlation coefficients between VEEV-TC-83 vRNA abundance and ~55,000 host genes. (D) Distributions of Spearman's correlation coefficients shown in D stratified by the average expression level of the gene in uninfected cells. N.S., not significant; (E) Representative genes with distinct expression patterns between uninfected, low vRNA and high vRNA cell groups. *, p < 0.05 by Mann–Whitney U test. (F) The expression of genes shown in (E) does not significantly change over time in uninfected cells. (TIF)

**S3 Fig. VEEV gap reads identified via viscRNA-Seq.** (A) Coverage of VEEV gap reads over the VEEV-TC-83-GFP genome. (B) Scatter plot of number of VEEV gap reads and VEEV total reads within cells with detected gap reads. Each dot represents a cell and colored by hpi. (C) Histogram of gap lengths indicating that the majority of gaps are shorter than 1000 nucleotides. (D) RNA structural prediction of the most common 36-base gap (arrow in A) via RNA-fold web server. Scale bar indicates the possibilities of base pairing. Hpi, hours postinfection. (TIF)

**S4 Fig. Loss-of-function and gain-of-function experiments for validation of candidate proviral and antiviral factors.** (A) Confirmation of gene expression knockdown in U-87 MG cells transfected with the indicated siRNAs or non-targeting control (NT) via qRT-PCR at 96 hours post-transfection. Results are relative to the level of the respective genes in the NT control. (B and E) Overall VEEV-TC-83 infection (grey) measured by luminescence assays and cell viability (orange) measured by alamarBlue assays in U-87 MG cells transfected with the indicated siRNAs (B) or ectopically expressing the indicated cellular factors (E) at 18 hpi with VEEV-TC-83-nLuc (MOI = 0.01). Data are expressed relative to siNT (B) or empty plasmid (E) controls. (C and F) Absolute fluorescence values from the alamarBlue assays shown in B and E. (D) Confirmation of ectopic expression of the indicated gene products tagged with a FLAG-tag by Western blot in U-87 MG cells. Membranes were blotted with anti-FLAG antibody. Samples in the left panels were run on the same gel from which several lanes were cut out. TAF7 expression on the right is shown at a higher exposure. Data sets are pooled from two independent experiments with six replicates each (B,C, E and F). Shown are means ± SD. (TIF)

**S5 Fig. Functional relevance of viscRNA-seq hits in cells infected with wild type TC-83 and TrD VEEV.** (A) Viral genome copies in lysates derived from U-87 MG cells transfected with the indicated siRNAs 24 hpi with non-reporter VEEV-TC-83 at an MOI of 0.01. (B and D) VEEV infection via plaque assays in U-87 MG cells transfected with the indicated siRNAs 24 hpi with non-reporter TC-83 (B) and TrD (D) (MOI = 0.001). C. Cell viability via alamarBlue assays in U-87 MG cells transfected with the indicated siRNAs. Shown are means ± SD. Data are plotted relative to non-targeting (NT) siRNA control (A, B and D). Representative experiments of at least two conducted are shown. *, q < 0.05; **, q < 0.01; *** q < 0.001 by 1-way ANOVA followed by False Discovery Rate (FDR) corrected multiple comparisons test. N.S, non-significant. (TIF)

**S6 Fig.** Pathway analysis for genes that positively correlated with VEEV 3'/5' read ratio and positively (A) or negatively (B) correlated with DENV and ZIKV. Each bar represents a group of genes according to Gene Ontology, KEGG, or other databases of biological function. The plot was made using metascape. (TIF)

**S1 Table. VEEV capture oligonucleotides.** (XLSX)

**S2 Table. siRNA sequence of candidate genes.** (XLS)

**S3 Table. qRT-PCR primer sequences.** (XLSX)

## Author Contributions

**Conceptualization:** Zhiyuan Yao, Fabio Zanini, Shirit Einav.

**Data curation:** Zhiyuan Yao, Fabio Zanini, Sathish Kumar, Marwah Karim, Sirle Saul, Nishank Bhalla, Nuttada Panpradist, Avery Muniz.

**Formal analysis:** Zhiyuan Yao, Fabio Zanini, Sathish Kumar, Marwah Karim, Sirle Saul, Nishank Bhalla, Nuttada Panpradist, Avery Muniz.

 

**Funding acquisition:** Aarthi Narayanan, Stephen R. Quake, Shirit Einav.

**Methodology:** Zhiyuan Yao, Fabio Zanini, Shirit Einav.

**Resources:** Aarthi Narayanan, Stephen R. Quake, Shirit Einav.

**Supervision:** Shirit Einav.

**Validation:** Zhiyuan Yao, Fabio Zanini, Sathish Kumar, Marwah Karim, Sirle Saul, Nishank Bhalla.

**Writing – original draft:** Zhiyuan Yao, Fabio Zanini, Shirit Einav.

**Writing – review & editing:** Marwah Karim, Sirle Saul, Shirit Einav.

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
