## [Decision Letter · Decision Letter 0]

8 Sep 2020

Dear Dr. Einav,

Thank you very much for submitting your manuscript "The transcriptional landscape of Venezuelan equine encephalitis virus (TC-83) infection" for consideration at PLOS Neglected Tropical Diseases. As with all papers reviewed by the journal, your manuscript was reviewed by members of the editorial board and by several independent reviewers. In light of the reviews (below this email), we would like to invite the resubmission of a significantly-revised version that takes into account the reviewers' comments. 

The manuscript has been reviewed by three experts in the field and, while the approach for single cell sequencing of VEEV-infected cells is considered novel and interesting, there were several criticisms of the experimental system. In particular, use of TC83 attenuated vaccine strain and a conventional immortalized cell line are considered to be poorly representative of the wild type virus and host cells in their natural context. It is recommended that the conclusions be significantly re-worked to account for the artificial in vivo system being used and the attenuated VEEV strain. This shroud integrate relevant literature on potential impacts of attenuating mutations in TC83 on host cell responses and citation of the appropriate literature. Also, the comments of reviewer 1 regarding additional experiments should be addressed.

We cannot make any decision about publication until we have seen the revised manuscript and your response to the reviewers' comments. Your revised manuscript is also likely to be sent to reviewers for further evaluation.

Sincerely,

William Klimstra

Associate Editor

Scott Weaver

Deputy Editor

The manuscript has been reviewed by three experts in the field and, while the approach for single cell sequencing of VEEV-infected cells is considered novel and interesting, there were several criticisms of the experimental system. In particular, use of TC83 attenuated vaccine strain and a conventional immortalized cell line are considered to be poorly representative of the wild type virus and host cells in their natural context. It is recommended that the conclusions be significantly re-worked to account for the artificial in vivo system being used and the attenuated VEEV strain. This shroud integrate relevant literature on potential impacts of attenuating mutations in TC83 on host cell responses and citation of the appropriate literature. Also, the comments of reviewer 1 regarding additional experiments should be addressed.

Reviewer's Responses to Questions

**Key Review Criteria Required for Acceptance?**

**Methods**

-Are the objectives of the study clearly articulated with a clear testable hypothesis stated?

-Is the study design appropriate to address the stated objectives?

-Is the population clearly described and appropriate for the hypothesis being tested?

-Is the sample size sufficient to ensure adequate power to address the hypothesis being tested?

-Were correct statistical analysis used to support conclusions?

-Are there concerns about ethical or regulatory requirements being met?

Reviewer #1: The study design was found to be acceptable. The hypothesis has been described and the sample sizes and statistical analyses appear appropriate. There are no concerns about ethical or regulatory requirements being met by the included studies.

Reviewer #2: Yes.

Reviewer #3: The goals and objectives of the study were very clearly stated. The approach used -- viscRNAseq, virus-inclusive single cell RNA-seq --is powerful and enabled the authors to ask very specific questions on virus-host interactions at a single cell level.

SMART-seq limits the number of cells that can be analyzed, but it is the best approach when capturing both vRNA and mRNA, and the number of cells is sufficient for the statistics performed.

All analyses were carefully done and the correct statistical analyses were chosen, supporting the conclusions.

**Results**

-Does the analysis presented match the analysis plan?

-Are the results clearly and completely presented?

-Are the figures (Tables, Images) of sufficient quality for clarity?

Reviewer #1: The analysis presented lines up with the authors' goals of identifying host transcripts that provide the basis for a proviral versus antiviral environment. The description of results are at times condensed and the readers may benefit from an expanded description. The clarity of the figures are acceptable.

Reviewer #2: Yes

Reviewer #3: (No Response)

**Conclusions**

-Are the conclusions supported by the data presented?

-Are the limitations of analysis clearly described?

-Do the authors discuss how these data can be helpful to advance our understanding of the topic under study?

-Is public health relevance addressed?

Reviewer #1: The conclusions in some cases appear to be too broad, beyond what is demonstrated by the provided datasets. The limitations of the methodology have not been well described. Further, the authors performed these experiments with the attenuated strain of the pathogen in a single cell type. As it is very evident from the subsequent analyses that they have performed with additional flaviviruses, the nature of the pathogen and the cell line can have enormous impacts on the conclusions, especially with regard to proviral and antiviral genes. The rationale for choosing flaviviruses for comparison with alphaviruses has not been described and the relevance to pathology related to alphaviruses is unclear.

Reviewer #2: Unfortunately the study design suffers from significant flaws and the conclusions/interpretation is limited.

Reviewer #3: (No Response)

**Editorial and Data Presentation Modifications?**

Reviewer #1: (No Response)

Reviewer #2: Yes

Reviewer #3: The work described in this study was performed with a live-attenuated vaccine strain – TC-83 because it also replicates rapidly. The discussion would benefit from a sentence or 2 on why this strain is representative of what we would expect with a fully virulent strain, and that the host genes identified are likely to be the same. This is meant for a more general audience who is not familiar with VEEV and its vaccine strain.

One aspect I would have liked the authors to maybe elaborate a bit more on, probably in the discussion, is the observation of gapped reads. These could indicate the presence of defective virus genomes, which are observed in many RNA viruses. There are a number of recent studies on defective viruses in other systems, including scRNAseq studies of influenza, and the potential of these defective genomes in interference. The authors indicate that they can’t determine if these gapped reads are an artifact of the polymerase during amplification but they could mention something about defective virus genomes not being an uncommon phenomenon. The authors do make a point to provide text and data in the supporting information on what they observe, so this is a minor point.

Line 391: The text in parenthesis should be reversed to show “(virus reads/total reads > 0.001)” and “(virus reads/total reads < 0.001)”. The way it is currently stated is confusing.

Fig 2A: there is a typo in the label pointing to the green circle -- “differntial”

Fig S1: Panel E GFP panel needs a label on the Y axis.

**Summary and General Comments**

Reviewer #1: This manuscript is based on the idea that transcriptomic changes are a reflection of a productive infection and can provide pertinent information that can help us understand the viral requirements to establish an infection. The authors have made excellent use of a single cell based transcriptomic analysis set up and drawn conclusions about host factors at the transcript level that impact the virus. The authors have used U87MG cells and infected them with TC-83. This is a well established model and there are several manuscripts that are published about transcriptomic changes, although not at the single cell level. While the authors allude to host factors that are published in the literature as being relevant to VEEV multiplication, they do not comment on whether these requirements are indeed at the level of the host transcript. Figure 1 would benefit from an experimental demonstration such as a PCR (a figure) that verifies the infectivity state prior to sample processing (Figure 1A, between the incubation and sample processing steps). In Figure 1C, the increase in the fraction of infected cells between the 6 and the 12 hour time points is too dramatic without a stepwise increase. It may be beneficial to include an additional time point between these two points. It is not clear what the authors refer to as mock-infected. Ideally, it should be some kind of a replication incompetent virus (UV-inactivated) and comparisons should be made in that context. When the authors identify certain pro and antiviral transcripts and attempt to do loss and gain of function assays, no data is included that shows the extent of depletion or overexpression. This is an important piece of information which will be critical to the determination of how much a given transcript actually matters to the virus replication. Absolute cell survival numbers without normalization should be shown for the depletions and the over expression contexts. The impact on the virus is based only on luciferase reporter viruses. The validation should be done in the context of TC-83 (without the reporter) and absolute genomic copy numbers and infectious titre counts should be obtained. More importantly, the relevance to disease cannot be ascertained unless the relevance of these transcripts are measured in the context of disease-causing virus (wild type VEEV).

Reviewer #2: The authors show transcriptional profile of a VEEV vaccine strain TC-83 in human cell line derived from malignant glioma. However, the experimental design suffers from significant issues that confound the interpretation of results.

Major issues:

1) Performing transcriptional profile of a virus infection in cell line derived from malignant glioma is in appropriate as the immortalized cells are by definition are not normal and important pathways such as antiviral response may be altered or absent. 

2) The VEE complex is comprised of 6 subtypes and 7 varieties. VEEV-IAB and IC are epidemic viruses, whereas all others are endemic viruses. The utilization of a vaccine strain that by definition is not a wild-type virus is inappropriate as the transcription profile may be vastly different than wild-type viruses. 

3) TC-83 attenuation is due to 2 point mutations; 5’ UTR nucleotide G3A and E2-120 Thr-to-Lys. The 5’ UTR point mutation is in a structural element that renders the vaccine sensitive to inhibition by Ifit1, whereas the wild-type virus resistant to inhibition by Ifit1. This host-pathogen interaction suggests that there may be considerable differences between the TC-83 and VEEV-IAB in transcriptional profiles.

4) The second attenuating point mutation in E2 of TC-83 virus is an adaptation to Heparan sulfate receptor. This adaptation enables rapid adsorption of virus in susceptible cell lines. This rapid adsorption in cells producing infectious virus particles will lead to re-adsorption of virus particles upon release. This effect may explain the “superproducer” effect and the subsequent transcriptional profile difference.

Reviewer #3: In this very interesting study, the authors set out to profile the dynamics of host and viral RNA (vRNA) in cells infected with VEEV using scRNAseq—and most specifically viscRNAseq, which is a method previously developed by this group for flaviviruses where they profile genomic viral RNA along with viral and host cell mRNA. VEEV is an alphavirus that naturally occurs in tropical areas of the world where it is transmitted via a mosquito vector; however, it is also a biothreat agent because of the fact it is also highly infectious as an aerosol.

The overall goal of this study is clearly stated and aims to get a better handle on the biology of this virus by identifying cell host factors that the virus highjacks to support its very high replication. The main questions that this study sets out to answer are (1) whether the high yield observed when the virus replicates rapidly is due to multiple cells being infected, or whether a few super producing cells are responsible; and (2) which host genes are proviral, supporting replication of the virus while the rest of host gene production is shut down. 

While scRNAseq studies have been done on other RNA viruses, this is the first to tackle VEEV. It also demonstrates the power of a method such as viscRNA-seq which allows the capture of both ends of the virus rather than being skewed to the poly-A tail end of the virus, providing a better handle on vRNA stochiometry measured by looking at 3’/5’ read coverage ratios, which reflect the ratio of structural to non-structural genes expressed. The method also enabled a unique comparative analysis of scRNAseq data across RNA viruses to determine what host gene expression profiles are shared during infection.

One intriguing and novel observation is that a small percentage of cells produced a very large number of viruses within 6 hours, which corresponds to a single cycle of VEEV replication. These superproducing cells were shown to have a characteristic transcriptional profile that distinguished them from other cells. This allowed the identification of putative proviral and antiviral factors. Rather than simply making that observation, the authors validated some of these candidates using loss-of-function screens by siRNA-mediated depletion and gain-of-function screens.

This is beautiful work, carefully executed, done in a stringent manner and the manuscript is clearly written. The authors do some clever analyses like stratifying their cells by vRNA abundance rather than by timepoint, and using the ERCC to normalize the host gene reads rather than using total reads.

PLOS authors have the option to publish the peer review history of their article (what does this mean?). If published, this will include your full peer review and any attached files.

Reviewer #1: No

Reviewer #2: No

Reviewer #3: No
---

## [Decision Letter · Decision Letter 1]

12 Mar 2021

Dear Dr. Einav,

We are pleased to inform you that your manuscript 'The transcriptional landscape of Venezuelan equine encephalitis virus (TC-83) infection' has been provisionally accepted for publication in PLOS Neglected Tropical Diseases. While one of the reviewers was unsure of the novelty of the studies, the other reviewer and the editors feel that the single cell sequencing components of the studies provide sufficient novelty for publication in the journal. 

Best regards,

William Klimstra

Associate Editor

Scott Weaver

Deputy Editor

Reviewer's Responses to Questions

**Key Review Criteria Required for Acceptance?**

**Methods**

-Are the objectives of the study clearly articulated with a clear testable hypothesis stated?

-Is the study design appropriate to address the stated objectives?

-Is the population clearly described and appropriate for the hypothesis being tested?

-Is the sample size sufficient to ensure adequate power to address the hypothesis being tested?

-Were correct statistical analysis used to support conclusions?

-Are there concerns about ethical or regulatory requirements being met?

Reviewer #2: The objectives are clearly stated.

Reviewer #3: This was the first study to use scRNAseq on VEE to capture both host cell and virus RNA. The application of viscRNAseq was clever, and the analyses were stringent.

**Results**

-Does the analysis presented match the analysis plan?

-Are the results clearly and completely presented?

-Are the figures (Tables, Images) of sufficient quality for clarity?

Reviewer #2: The results are clearly presented.

Reviewer #3: The results were well presented and the authors addressed appropriately all the comments.

**Conclusions**

-Are the conclusions supported by the data presented?

-Are the limitations of analysis clearly described?

-Do the authors discuss how these data can be helpful to advance our understanding of the topic under study?

-Is public health relevance addressed?

Reviewer #2: The conclusions are supported by the data.

Reviewer #3: (No Response)

**Editorial and Data Presentation Modifications?**

Reviewer #2: N/A

Reviewer #3: (No Response)

**Summary and General Comments**

Reviewer #2: The current manuscript investigated transcriptional profile of TC-83 in three different immortalized cell lines. The study utilizes cell lines with defects and/or alterations in host response pathways. The study is not novel and the latter is a considerable flaw. Multiple previous studies have investigated TC-83 and VEEV IAB gene expression in relevant animal models and human samples from TC-83 vaccine recipients (see below). Consequently, the data from previous studies is more pertinent to profiling host response to TC-83 and VEEV IAB infection. This manuscript appears to be a method development study and should be submitted to a more technical journal.

Hammamieh R, Barmada M, Ludwig G, Peel S, Koterski N, Jett M. Blood genomic profiles of exposure to Venezuelan equine encephalitis in Cynomolgus hammamiehmacaques (Macaca fascicularis). Virology J 2007; 4:82.

Koterski J, Twenhafel N, Porter A, Reed DS, Martino-Catt S, Sobral B, Crasta O, Downey T, DaSilva L. Gene expression profiling of nonhuman primates exposed to aerosolized Venezuelan equine encephalitis virus. FEMS Immunol Med Microbiol 2007; 51(3):462-72.

Sharma A, Bhattacharya B, Puri RK, Maheshwari RK. Venezuelan equine encephalitis virus infection causes modulation of inflammatory and immune response genes in mouse brain. BMC Genomics 2008; 9:289.

Sharma A, Maheshwari RK. Oligonucleotide array analysis of Toll-like receptors and associated signaling genes in Venezuelan equine encephalitis virus-infected mouse brain. J Gen Virol 2009; 90:1836-47.

Sharma A, Bhomia M, Honnold SP, Maheshwari RK. Role of adhesion molecules and inflammation in Venezuelan equine encephalitis virus infected mouse brain. Virology J 2011; 8:197.

Bhomia M, Balakathiresan N, Sharma A, Gupta P, Biswas R, Maheshwari RK. Analysis of microRNAs induced by Venezuelan equine encephalitis virus infection in mouse brain. BBRC 2010; 395:11-16.

Erwin-Cohen RA, Porter A, Pittman PR, Rossi CA, DaSilva L. (2012). Host responses to live-attenuated Venezuelan equine encephalitis virus (TC-83): Comparison of naïve, vaccine responder and NonResponder to TC-83 challenge in human peripheral blood mononuclear cells. Hum Vaccin Immunother 2012; 8(8):1053-65;

Gupta P, Sharma A, Han J, Yang A, Bhomia M, Knollmann-Ritschel B, et al. Differential host gene responses from infection with neurovirulent and partially-neurovirulent strains of Venezuelan equine encephalitis virus. BMC Infect Dis. 2017 Apr 26;17(1):309.

Erwin-Cohen RA, Porter AI, Pittman PR, Rossi CA, DaSilva L. Human transcriptome response to immunization with live-attenuated Venezuelan equine encephalitis virus vaccine (TC-83): Analysis of whole blood. Hum Vaccin Immunother. 2017 Jan 2;13(1):169-179.

Reviewer #3: (No Response)

PLOS authors have the option to publish the peer review history of their article (what does this mean?). If published, this will include your full peer review and any attached files.

Reviewer #2: No

Reviewer #3: No

---

## [Editor Report · Acceptance letter]

25 Mar 2021

Dear Dr. Einav,

We are delighted to inform you that your manuscript, " The transcriptional landscape of Venezuelan equine encephalitis virus (TC-83) infection ," has been formally accepted for publication in PLOS Neglected Tropical Diseases.

Best regards,

Shaden Kamhawi

co-Editor-in-Chief

Paul Brindley

co-Editor-in-Chief
